# A Unified Total Variation Framework for Membrane Potential Perturbation Dynamic

**Zhao-Rong Lai**[1], **Xiwen Yuan**[2*], **Ziliang Chen**[3], **Liangda Fang**[4], **Yongsen Zheng**[5,6]

[1]Guangdong Key Laboratory of Data Security and Privacy Preserving,
  College of Cyber Security, Jinan University
[2]Department of Mathematics,
  College of Information Science and Technology, Jinan University
[3]Research Institute of Multiple Agents and Embodied Intelligence, Peng Cheng Laboratory
[4]Department of Computer Science,
  College of Information Science and Technology, Jinan University
[5]College of Computing and Data Science, Nanyang Technological University
[6]Digital Trust Centre, Nanyang Technological University
`laizhr@jnu.edu.cn, yxw20041225@stu2023.jnu.edu.cn`
`c.ziliang@yahoo.com, fangld@jnu.edu.cn, yongsen.zheng@ntu.edu.sg`

## Abstract

Membrane potential perturbation dynamic (MPPD) is an emerging approach to capture perturbation intensity and stabilize the performance of spiking neural networks (SNN). It discards the neuronal reset part to intuitively reduce fluctuations of dynamics, but this treatment may be insufficient in perturbation characterization. In this study, we prove that MPPD is total variation (TV), which is a widely-used methodology for robust signal reconstruction. Moreover, we propose a novel TV-$\ell_1$ framework for MPPD, which allows for a wider range of network functions and has better denoising advantage than the existing TV-$\ell_2$ framework, based on the coarea formula. Experiments show that MPPD-TV-$\ell_1$ achieves robust performance in both Gaussian noise training and adversarial training for image classification tasks. This finding may provide a new insight into the essence of perturbation characterization.

## 1 Introduction

Spiking neural networks (SNN, Maass 1997) are a main category of NNs that have caught more and more attention these years (Shen et al., 2023; Li et al., 2024; Song et al., 2024). Since they have sparse activation features (Yao et al., 2025), they require less computational complexity and power in training and operating than the NNs with dense activation features (Fang et al., 2023), making it an advantage in deep learning scenarios (Pei et al., 2019; Perez-Nieves & Goodman, 2021). A key concept to convey binary information in the SNN is the *membrane potential* (Xu et al., 2023; Zhu et al., 2024; Ding et al., 2024a), which imitates the complex dynamics in the brain (Zhang & Li, 2020; Shi et al., 2024b; Yao et al., 2022). Such a concept bridges the computational properties of the SNN with those of the biological neural system, which opens up a promising research topic for future works.

Similar to other categories of NNs, SNNs are vulnerable to attacks from adversarial examples (Goodfellow et al., 2015; Kundu et al., 2021; Ding et al., 2022; Bu et al., 2023; Hao et al., 2024). It holds back applications of SNNs to scenarios with strict security needs (Yamazaki et al., 2022; Liang et al., 2023; Wu et al., 2024; Sharmin et al., 2019; Ding et al., 2024a;b; Geng & Li, 2025). One solution to this problem is to effectively identify the adversarial perturbations. By observing that membrane potentials contain adversarial perturbation information in the leaky integrate-and-fire (LIF) neuron based SNNs (Sharmin et al., 2020), a kind of membrane potential perturbation dynamics (MPPD) is proposed to analyze the dynamic properties of such perturbation information (Ding et al., 2024a). It further proposes to use the mean square of MPPD (MS-MPPD) as a regularizer to stabilize the performance of SNNs against adversarial examples.

---
*Corresponding author.

To better highlight our motivation, we first provide the formula of MPPD (Ding et al., 2024a):

$$\underbrace{\vartheta_i^l[t]}_{full\ MPPD} = \underbrace{\lambda\vartheta_i^l[t-1] + \sum_j w_{ij}^l(s_j^{l-1}[t] - \tilde{s}_j^{l-1}[t])}_{MPPD} - \underbrace{\lambda(v_i^l[t-1]s_i^l[t-1] - \tilde{v}_i^l[t-1]\tilde{s}_i^l[t-1])}_{neuronal\ reset}, \tag{1}$$

where $v_i^l[t]$ and $s_i^l[t]$ denote the *pure* membrane potential and spike of neuron $i$ in layer $l$ at time $t$, respectively. The notation with the *tilde* superscript is the *perturbed* version of the corresponding variable. The *full MPPD* is defined as the difference between the pure and the perturbed membrane potential: $\vartheta_i^l[t] = v_i^l[t] - \tilde{v}_i^l[t]$, which equals the *MPPD* part on the right side of (1) when this neuron does not fire a spike. However, if the neuron fires a spike, there will be an additional *neuronal reset* part, which might cause fluctuations in $\vartheta_i^l[t]$. Hence this neuronal reset part is discarded by (Ding et al., 2024a) and only the MPPD part is used. Though intuitive, this treatment may be insufficient in perturbation characterization.

In this study, we discover and prove that MPPD is total variation (TV), which is a widely-used methodology in robust signal reconstruction (Rudin et al., 1992; Chan et al., 2006; Chen et al., 2006), function approximation (Chan & Esedoglu, 2005), and invariant risk minimization (Lai & Wang, 2024; Wang et al., 2025). TV accumulates the increments of a function with respect to (w.r.t.) its arguments, which well fits the perturbation of membrane potential. The proposed methodology requires only one fundamental condition that the perturbation is a measurable function of the node index and the time-step of an SNN. This means that the temporal and neuronal evolutions of the SNN should be able to capture such a perturbation and then yield significant TV. This finding may provide a new insight into the essence of characterizing perturbations.

Our main contributions can be summarized as follows: **1.** We prove that MPPD is TV and verify that the existing MS-MPPD regularized SNN training model is a standard TV-$\ell_2$ framework. **2.** We further propose a novel TV-$\ell_1$ framework for MPPD (MPPD-TV-$\ell_1$). It has at least two advantages over the TV-$\ell_2$ framework: a) The $L^1$ function space is larger than the $L^2$ function space in general deep learning settings, which allows for more classes of functions to be membrane potentials. b) Based on the coarea formula, TV-$\ell_1$ performs better than TV-$\ell_2$ in robust signal reconstruction against adversarial perturbations, which better fits the architectures of SNNs. **3.** We deduce the coarea formula, the dominated TV property, and the subgradient calculation for MPPD-TV-$\ell_1$. Our methodology is applicable to most SNN architectures where a TV term is used to stabilize layer-wise internal state.

## 2 PRELIMINARIES AND RELATED WORKS

We introduce some preliminaries and related works on SNNs, MPPD, TV, and adversarial attacks.

### 2.1 MEMBRANE POTENTIAL PERTURBATION DYNAMICS

Different from typical analog neural networks (ANNs), SNNs use spike sequences to send temporal binary information. This mechanism imitates the dynamic communications of biological neural systems. To exploit temporal spike information, the LIF model (Wu et al., 2019; Kim et al., 2022; Shi et al., 2024a) can be used to characterize how neurons work in SNNs. The discrete-form differential equations of LIF are as follows:

$$v_i^l[t] = \lambda u_i^l[t-1] + \sum_j w_{ij}^l s_j^{l-1}[t], \ s_i^l[t] = H(v_i^l[t] - u_{th}), \ u_i^l[t] = v_i^l[t](1 - s_i^l[t]), \tag{2}$$

where $v_i^l[t]$ denotes the membrane potential of neuron $i$ in layer $l$ at time-step $t$ before firing ($i \in [N^l] := \{1, 2, \cdots, N^l\}; t \in [T]; l \in [L]; u_i^l[0] = 0$). $H$ is the Heaviside function such that if $v_i^l[t]$ is no less than the threshold $u_{th}$, then the spike function $s_i^l[t] = 1$; else $s_i^l[t] = 0$. $w_{ij}^l$ denotes the weight of the edge connecting neurons $i$ and $j$, where $j$ is from the preceding layer $(l - 1)$. $u_i^l[t]$ denotes the membrane potential after firing, which resumes to the resting potential (i.e., 0) and waits for being transferred to the succeeding time-step with a decaying leaky factor $\lambda \in [0, 1)$.

As introduced in Section 1 and (1), the MPPD is defined as (Ding et al., 2024a):

$$\epsilon_i^l[t] = \lambda\epsilon_i^l[t-1] + \sum_j w_{ij}^l \Delta s_j^{l-1}[t], \quad t = 1, 2, \cdots, T, \tag{3}$$

where $\Delta s_j^{l-1}[t] := s_j^{l-1}[t] - \tilde{s}_j^{l-1}[t]$ denotes the difference of the presynaptic spike, and the neuronal reset part in (1) is discarded. Then the MS-MPPD regularized SNN training model is (Ding et al., 2024a):

$$\min_w \ \{\mathcal{L} := \mathcal{L}_{task} + \alpha \cdot \mathcal{L}_{MS\text{-}MPPD}\} \tag{4}$$

$$\text{s.t. } \mathcal{L}_{task} := \chi CE(f_{SNN}(x), y) + (1-\chi)CE(f_{SNN}(\tilde{x}), y), \mathcal{L}_{MS\text{-}MPPD} := \sum_{i=1}^{N^L} \sum_{t=1}^{T} (\epsilon_i^L[t])^2, \quad (5)$$

where $x \in \mathbb{R}^d$ (or $\tilde{x}$) and $y \in \mathbb{Y}$ denote the pure (or perturbed) input and the corresponding ground-truth output of the training samples, respectively. $f_{SNN}$ denotes the function induced by the SNN, and $CE$ denotes the cross-entropy classifier. $\chi$ is a mixing hyperparameter with a default value of 0.5, and $\alpha \geqslant 0$ is a regularization hyperparameter for $\mathcal{L}_{MS\text{-}MPPD}$. $N^L$ denotes the number of neurons in the $L$-th layer (the last layer). To summarize, model (4) sums up the mean squared perturbations with a factor $\alpha$ to the task loss, in order to suppress such perturbations in the training process. The training algorithm is a standard Spatio-Temporal BackproPagation (STBP) with the triangle-like surrogate function (Deng et al., 2022) in place of the non-differentiable Heaviside function with $\omega = 1$ by default:

$$\frac{\partial s_i^l[t]}{\partial v_i^l[t]} = \frac{1}{\omega^2} \max(\omega - |v_i^l[t] - u_{th}|, 0). \quad (6)$$

## 2.2 ADVERSARIAL ATTACKS

It is widely-recognized that NNs are vulnerable to adversarial attacks. These attacks deliberately change the input data for a little bit, in order to make the NNs give incorrect results. The SNN also suffers from this problem, although it has a higher activation sparsity than the ANN. A common attack strategy is to maximize the network loss when the classifier $f : \mathbb{R}^d \mapsto \mathbb{Y}$ makes an incorrect classification after receiving a perturbed input $(x + \delta)$:

$$\max_{\|\delta\|_p \leqslant \zeta} \mathcal{L}_{task}(f(x+\delta), y), \quad (7)$$

where $\|\delta\|_p \leqslant \zeta$ means that the attack is imperceptible with $\ell_p$ norm no more than an intensity hyperparameter $\zeta \geqslant 0$. For images, $\zeta$ is often set as an integer multiplied by $1/255$.

Fast Gradient Sign Method (FGSM, Goodfellow et al. 2015) is a fundamental attack technique to generate adversarial examples by adding a small component in the same direction of the gradient sign:

$$\tilde{x} = x + \zeta' \cdot \text{sign}(\nabla_x \mathcal{L}_{task}(f(x), y)), \quad (8)$$

where $0 \leqslant \zeta' \leqslant \zeta / \|\text{sign}(\nabla_x \mathcal{L}_{task}(f(x), y))\|_p$. Based on FGSM, the Projected Gradient Descent (PGD, Madry et al. 2018) attack technique is further proposed to achieve a stronger attack of "first-order adversary":

$$\tilde{x}_{(k+1)} = \text{proj}_{\mathcal{B}_p[x;\zeta]}(\tilde{x}_{(k)} + \eta \cdot \text{sign}(\nabla_x \mathcal{L}_{task}(f(\tilde{x}_{(k)}), y))), \quad (9)$$

where $\eta$ represents the step size for a single PGD iteration, and $\text{proj}_{\mathcal{B}_p[x;\zeta]}$ denotes the projection operator onto the closed neighborhood of $x$ with a radius of $\zeta$. Besides the above gradient-based attack schemes, C&W is also a popular optimization-based attack mechanism (Carlini & Wagner, 2017).

## 2.3 TOTAL VARIATION

TV (Rudin et al., 1992) is a widely-used operator that measures the degree of variation in a function. We consider an open set $\Omega \subseteq \mathbb{R}^d$ and denote $L^1(\Omega)$ as the corresponding Lebesgue-integrable function space. The TV of any $f \in L^1(\Omega)$ is originally defined as (Chan et al., 2006):

$$\int_\Omega |\nabla f| := \sup\{\int_\Omega f(x)\text{div}g(x)\,\mathrm{d}x : g \in C_c^1(\Omega, \mathbb{R}^d), \|g\|_{L^\infty(\Omega)} \leqslant 1\}, \quad (10)$$

where $\text{div}g$ denotes the divergence of a differentiable function $g$. $g$ has a compact support contained in $\Omega$ and an essential supremum no larger than 1. TV quantifies the local variations of $f$ across all dimensions of $x$, and then accumulates these variations over the entire domain $\Omega$. If $f$ is non-differentiable, then $|\nabla f|$ is characterized via $\text{div}g$. But when $f$ is differentiable, $|\nabla f|$ is exactly the magnitude (i.e., the $\ell_2$ norm) of the gradient $\nabla f$.

In practice, only functions with bounded variation (BV) can be calculated, which constitute the BV function space $\{f \in L^1(\Omega) : \int_\Omega |\nabla f| < \infty\}$. Hence a TV function actually refers to a BV function in this paper. TV has an important property that can be represented by the following coarea formula (Chan et al., 2006):

$$\int_\Omega |\nabla f| := \int_{-\infty}^{\infty} \int_{f^{-1}(\psi)} \mathrm{d}\varphi\,\mathrm{d}\psi, \quad (11)$$

where $f^{-1}(\psi) := \{x \in \Omega : f(x) = \psi\}$ represents the level set (or preimage) of $f$ at the value $\psi$. (11) indicates that this integral is calculated by aggregating all contours of $f^{-1}(\psi)$ for every $\psi$ where the differential $\mathrm{d}\psi$ exists. Hence if $f$ has a more blocky (piecewise-constant) landscape, its TV will be smaller. Based on this property, minimizing TV helps to preserve discontinuous features of $f$, while effectively reducing noise and other unwanted fine-scale details. The TV-$\ell_1$ model (Rudin et al., 1992) is proposed to this end:

$$\inf_{\hat{f} \in L^2(\Omega)} \left\{ \int_{\Omega} |\nabla \hat{f}| + \lambda \int_{\Omega} (f - \hat{f})^2 \, \mathrm{d}x \right\}, \tag{12}$$

where $\hat{f} \in L^2(\Omega)$ denotes the recovery of the target function $f$, and $\lambda$ is a hyperparameter that controls the approximation accuracy. This model aims to preserve the sharp discontinuities in the recovery $\hat{f}$ while using the residual $(f - \hat{f})$ to capture and retain noise as well as other unwanted fine-scale details.

If $\hat{f}$ has a stronger TV condition that $\int_{\Omega} |\nabla \tilde{f}|^2 < \infty$, then the following TV-$\ell_2$ framework can also be used (Mumford & Shah, 1985):

$$\inf_{\hat{f} \in L^2(\Omega)} \left\{ \int_{\Omega} |\nabla \hat{f}|^2 + \lambda \int_{\Omega} (f - \hat{f})^2 \, \mathrm{d}x \right\}. \tag{13}$$

Compared with TV-$\ell_1$, TV-$\ell_2$ uses the squared TV term $\int_{\Omega} |\nabla \hat{f}|^2$, which does not have the coarea formula (11) and thus lacks robustness to sharp noises.

## 3 METHODOLOGY

We observe that MPPD accords with the concept of TV not only by its formulation of (3) but also by its motivation to suppress perturbations. Hence we aim to establish a complete theory and methodology for the TV based MPPD, in order to improve its robustness to adversarial perturbations.

### 3.1 TOTAL VARIATION FORMULATION OF MEMBRANE POTENTIAL PERTURBATION DYNAMICS

(3) has a natural form of differences in both dimensions of time-step and node index. In the time-step dimension, the perturbation term $\epsilon_i^l[t]$ is influenced by its one-step-forward term $\epsilon_i^l[t-1]$. In the node index dimension, $\epsilon_i^l[t]$ is influenced by its preceding nodes $\sum_j w_{ij}^l \Delta s_j^{l-1}[t]$. Since the preceding nodes of a given node $i$ are fixed, we can omit the layer notation $l$ to simplify expressions. Then we need to verify that both sides of (3) are well-defined in the framework of TV.

**Part (a):** We first examine the left side of (3). By taking the node index $i$, the time-step $t$, and the input $x$ as arguments, the perturbation term intends to approximate the difference between the pure and the perturbed membrane potential:

$$\epsilon(i, t, x) \approx v(i, t, x) - v(i, t, \tilde{x}) = v(i, t, x) - v(i, t, x + \delta), \tag{14}$$

where $\delta$ denotes the perturbation added to the input $x$. We find that if $\delta$ can be represented by $i$ and $t$: $\delta := \delta(i, t)$, then the right side of (14) is exactly local variation of $v$ w.r.t. $(i, t)$ at $x$, which can be directly defined as $\epsilon(i, t, x)$:

$$\epsilon(i, t, x) := \nabla_{(i,t)} v(i, t, x) := v(i, t, x) - v(i, t, x + \delta(i, t)). \tag{15}$$

The reason is that the perturbation $\delta(i, t)$ can be embedded into the SNN at node $i$ and time-step $t$. The symbol $\nabla_{(i,t)}$ means that this difference is performed in the unified dimensions $(i, t)$ via $\delta(i, t)$. Then $\mathcal{L}_{MS\text{-}MPPD}$ in (5) is exactly a TV-$\ell_2$ term:

$$\int_1^{N^L} \int_1^T \epsilon^2(i, t, x) \, \mathrm{d}t \, \mathrm{d}i = \int_1^{N^L} \int_1^T |\nabla_{(i,t)} v(i, t, x)|^2 \, \mathrm{d}t \, \mathrm{d}i, \quad \forall x \in \mathbb{R}^d, \tag{16}$$

where the integrations w.r.t. $i$ and $t$ take the discrete form for a discrete-time SNN.

**Part (b):** The right side of (3) actually re-calculates the local variation of $v$ by exploiting the information flow based on the network structure. Similar to (15), the local variation of the spike function $s$ can be represented by:

$$\Delta s(j, t, x) = s(j, t, x) - s(j, t, x + \delta(i, t)) =: \nabla_{(j,t)} s(j, t, x). \tag{17}$$

Then the right side of (3) can be transformed into the following local variation:

$$\lambda \nabla_{(i,t)} v(i, t-1, x) + \int_{\mathcal{J}(i)} \nabla_{(j,t)} s(j, t, x) \, \mathrm{d}w(i, j(i)), \tag{18}$$

where $\mathcal{J}(i)$ denotes the index set of the nodes preceding $i$, and the weight $w(i, j(i))$ serves as a measure for the integral. $w(i, j(i))$ is assumed to be a finite measure in this paper, which is the case in general practice of neural networks. To simplify notations, we use $\int_{\mathcal{J}(i)} \mathrm{d}w(i, j(i))$ for the univariate integral w.r.t. $j$ with fixed $i$ and $\int_1^N \int_{\mathcal{J}(i)} \mathrm{d}w(i, j(i))$ for the bivariate integral w.r.t. the entire $(i, j(i))$, respectively. After integrating on $j$, the second term of (18) is still local variation at $(i, t, x)$. By joining both sides of (3), we can directly well define MPPD in the form of TV without entailing the neuronal reset part in (1). By this means, the perturbation $\delta$ can be fully conveyed by the MPPD throughout different nodes and time-steps of an SNN.

**Theorem 1.** *If the perturbation $\delta$ is a measurable function of $(i, t)$, then the following equations on local variation and TV hold:*

$$\nabla_{(i,t)} v(i, t, x) = \lambda \nabla_{(i,t)} v(i, t-1, x) + \int_{\mathcal{J}(i)} \nabla_{(j,t)} s(j, t, x) \, \mathrm{d}w(i, j(i)), \quad \forall (i, t, x), \quad (19)$$

$$\int_1^{N^L} \int_1^T |\nabla_{(i,t)} v(i, t, x)|^2 \, \mathrm{d}t \, \mathrm{d}i$$

$$= \int_1^{N^L} \int_1^T \left| \lambda \nabla_{(i,t)} v(i, t-1, x) + \int_{\mathcal{J}(i)} \nabla_{(j,t)} s(j, t, x) \, \mathrm{d}w(i, j(i)) \right|^2 \, \mathrm{d}t \, \mathrm{d}i, \quad \forall x. \quad (20)$$

*The above integrals allow for any feasible measure types for $i$ and $t$ in the mathematical form, including both discrete and continuous measures.*

The proof is provided in Appendix A.1. This appendix also verifies that the integral on the right side of (19) is finite, which is necessary for the dominated TV property of Theorem 4, in order to control the overall stability of an SNN. (20) is the TV-$\ell_2$ version of $\mathcal{L}_{MS\text{-}MPPD}$, denoted by MPPD-TV-$\ell_2$. This MPPD-TV-$\ell_2$ term is finite in general situations, as stated in Theorem 4. Therefore, adding $\alpha \cdot \mathcal{L}_{MS\text{-}MPPD}$ in (4) actually suppresses the squared TV of membrane potential throughout the entire SNN, in order to suppress the adversarial perturbations inside the TV. To do this, the condition that $\delta$ is measurable of $(i, t)$ is fundamental, otherwise $\delta$ cannot be fully identified by the SNN and yield significant TV. An intuitive interpretation of the term measurable is that different magnitudes of $\delta$ can be discriminated via different combinations of nodes and time-steps. This accords with the common sense that using more nodes and time-steps may improve the accuracy of identifying adversarial perturbations.

## 3.2 MPPD-TV-$\ell_1$

In the previous subsection, we prove that MPPD is TV and the corresponding $\mathcal{L}_{MS\text{-}MPPD}$ is equivalent to a TV-$\ell_2$ model. Without loss of generality, we can assume $\nabla_{(i,t)} v(i, 0, x) = 0$ and $t$ being an integer. Then (19) can be aggregated w.r.t. all the $k < t$ as follows:

$$\nabla_{(i,t)} v(i, t, x) = \sum_{k=0}^{t-1} \lambda^k \int_{\mathcal{J}(i)} \nabla_{(j,t)} s(j, t-k, x) \, \mathrm{d}w(i, j(i)), \quad (21)$$

which reveals a difference evolution process along the node $(i)$ and the time-step $(k)$ dimensions. Besides, $\nabla_{(i,t)} v(i, t, x)$ is directly the sum of spike perturbations, thus its absolute value (instead of the squared value) quantifies the exact magnitude of these perturbations. Moreover, TV-$\ell_1$ has at least two advantages over TV-$\ell_2$: 1) TV-$\ell_1$ can exploit the coarea formula to suppress adversarial perturbations. 2) With a finite measure, an $L^2$ integrable function is also an $L^1$ integrable function, but the converse is not true (see Appendix A.3). Hence the $L^1$ function space is larger than the $L^2$ function space with finite measures for $i$ and $t$ (for example, when $i \in [N]$, $t \in [T]$, and the counting measure is used), which allows for more classes of functions to be membrane potentials, and expands the applicability and flexibility of TV-$\ell_1$. These findings and advantages motivate us to develop a novel TV-$\ell_1$ framework for MPPD (MPPD-TV-$\ell_1$).

The first step is to establish the coarea formula for the membrane potential $v$. We denote the unified domain of $(i, t)$ by $\Theta$ and the corresponding measure by $\mu$. For example, $\Theta = [N] \times [T]$ and $\mu(\{(i, t)\}) = 1, \forall (i, t) \in \Theta$ can be used for a standard discrete SNN.

**Theorem 2** (Coarea Formula). *If the perturbation $\delta$ is a measurable function of $(i,t)$, the following coarea formula holds for both continuous and discrete settings:*

$$\int_\Theta |\nabla_{(i,t)} v(i,t,x)| \, \mathrm{d}\mu = \int_{-\infty}^\infty \int_{\{(i,t)\in\Theta : v(i,t,x)=\psi\}} \mathrm{d}\varphi \, \mathrm{d}\psi, \quad \forall x, \tag{22}$$

*where $\varphi$ denotes the Hausdorff measure induced by $\mu$.*

The proof is provided in Appendix A.2. Taking the above standard discrete SNN as an example, the coarea formula counts the number of points $(i,t)$ at which $v(\cdot,\cdot,x)$ equals a fixed $\psi$, then aggregates all the infinitesimal surface areas along $\psi \in (-\infty,\infty)$: $\sum_\psi \varphi(\{(i,t)\in\Theta : v(i,t,x)=\psi\}) \cdot \Delta\psi$. In brief, the TV will increase significantly if the Hausdorff measure $\varphi(\{(i,t)\in\Theta : v(i,t,x)=\psi\})$ corresponding to the interval $[\psi, \psi+\Delta\psi)$ is large. Such intervals and points may contain target perturbations and thus could be suppressed in the objective. Based on this property, we can develop the MPPD-TV-$\ell_1$ framework as follows.

**Theorem 3.** *If the perturbation $\delta$ is a measurable function of $(i,t)$, then the following MPPD-TV-$\ell_1$ is well-defined:*

$$\int_\Theta |\nabla_{(i,t)} v(i,t,x)| \, \mathrm{d}\mu = \int_\Theta \left| \sum_{k=0}^{t-1} \lambda^k \int_{\mathcal{J}(i)} \nabla_{(j,t)} s(j, t-k, x) \, \mathrm{d}w(i, j(i)) \right| \mathrm{d}\mu, \quad \forall x. \tag{23}$$

*The above integrals allow for any feasible measure types $\mu$ in the mathematical form, including both discrete and continuous measures.*

The proof is provided in Appendix A.3. This TV formulation penalizes the total accumulation of potential changes over time, not just the potential at the moment of a spike. The membrane potential $v(i,t,x)$ evolves continuously based on input currents, even when no spike occurs. Replacing the $\mathcal{L}_{MS\text{-}MPPD}$ term in (4) by the MPPD-TV-$\ell_1$ term in (23), the new MPPD-TV-$\ell_1$ framework can be used for different tasks.

### 3.3 Properties of MPPD-TV-$\ell_1$

We investigate two useful properties of MPPD-TV-$\ell_1$ in this subsection. First, an important property of SNN is that the perturbation dynamics should be dominated by the spikes (Khalil, 2002). We verify that this property also holds in both MPPD-TV-$\ell_1$ and MPPD-TV-$\ell_2$ frameworks.

**Theorem 4** (Dominated TV Property). *To simplify notation, assume every node $i$ in layer $l$ uses the same set of preceding nodes $\mathcal{J}$ in layer $(l-1)$. Then for MPPD-TV-$\ell_1$ defined in (21) and (23) and MPPD-TV-$\ell_2$ defined in (20), the following dominated TV property holds:*

$$\int_1^{N^l} \int_1^T |\nabla_{(i,t)} v(i,t,x)| \, \mathrm{d}t \, \mathrm{d}i \leqslant \|w_l\|_1 \log_\lambda\left(\frac{1}{e}\right) \int_{\mathcal{J}} \int_1^T |\nabla_{(j,t)} s(j,t,x)| \, \mathrm{d}t \, \mathrm{d}j, \tag{24}$$

$$\int_1^{N^l} \int_1^T |\nabla_{(i,t)} v(i,t,x)|^2 \, \mathrm{d}t \, \mathrm{d}i \leqslant \|w_l\|_F^2 \log_\lambda^2\left(\frac{1}{e}\right) \int_{\mathcal{J}} \int_1^T |\nabla_{(j,t)} s(j,t,x)|^2 \, \mathrm{d}t \, \mathrm{d}j, \quad \forall l, \forall x, \tag{25}$$

*where $\|w_l\|_1$ and $\|w_l\|_F$ denote the 1-norm and the Frobenius norm of the weight matrix $w_l$ connecting layers $l$ and $(l-1)$, respectively. (24) also holds in the discrete form by replacing the scaling factor $\|w_l\|_1 \log_\lambda(\frac{1}{e})$ by $\frac{\|w_l\|_1}{1-\lambda}$.*

The proof is provided in Appendix A.4. For general situations such as sparse or skip-connected SNN architectures, the weight matrix $w_l$ simply has zero entries corresponding to non-connections, and the set of preceding nodes $\mathcal{J}(i)$ is a proper subset of the entire layer $(l-1)$. The proof structure and the fundamental dominating bound remain valid. Note that $|\nabla_{(j,t)} s(j,t,x)| \leqslant 1$ from the definition of the Heaviside function, and the integration limits for $t$ and $j$ are also finite. Hence the integral on the right side of (24) or (25) is finite, which is able to control the left side. This theorem indicates that the TV of membrane potential $v$ is dominated by the TV of spike $s$ up to a factor of $\|w_l\|_1 \log_\lambda(\frac{1}{e})$. In this factor, $\|w_l\|_1$ indicates the spectral energy spread caused by the edge weight, while $\log_\lambda(\frac{1}{e})$ indicates the scaling effect caused by temporal evolution. The closer $\lambda$ is set to 1, the larger scaling of spike TV is required to dominate the membrane potential TV. Nevertheless, $\lambda$ is

usually set very close to 1 to simultaneously keep the smoothness of temporal evolution and ensure the above dominated TV property.

The right side of (23) is nondifferentiable w.r.t. the weight $w(i, j(i))$, thus $w(i, j(i))$ cannot be trained by mainstream learning architectures like Pytorch[1]. To solve this problem, we complete the backpropagation module with a closed-form subgradient calculation of (23). This strategy is widely-used to deal with nondifferentiable terms in general machine learning tasks (Lin et al., 2024a;b).

**Proposition 5** (Subgradient Calculation). *A subgradient of* (23) *w.r.t. the weight* $w(i, j(i))$ *can be calculated as follows:*

$$\int_{\Theta} \partial_{w(i,j(i))} \left| \sum_{k=0}^{t-1} \lambda^k \int_{\mathcal{J}(i)} \nabla_{(j,t)} s(j, t-k, x) \, \mathrm{d}w(i, j(i)) \right| \mathrm{d}\mu$$

$$= \int_{\Theta} \mathrm{sign}(\sum_{k=0}^{t-1} \lambda^k \int_{\mathcal{J}(i)} \nabla_{(j,t)} s(j, t-k, x) \, \mathrm{d}w(i, j(i))) \cdot (\sum_{k=0}^{t-1} \lambda^k \nabla_{(j,t)} s(j, t-k, x)) \, \mathrm{d}\mu. \quad (26)$$

The proof is provided in Appendix A.5. It can be seen that this subgradient calculation works as a standard gradient calculation and will not cause additional computational complexity. Based on this property, the MPPD-TV-$\ell_1$ framework is compatible with mainstream learning architectures and enables the training of SNNs. Moreover, this subgradient captures the sensitivity of the TV to the weights at every timestep $t$, regardless of whether the potential crosses the threshold and is reset. By minimizing the TV, the model weights are enforced to produce membrane potential trajectories that are globally smoother and less responsive to small changes in the input (noise). This inherent smoothness regulates the weight updates such that even small, sub-threshold input perturbations are suppressed by a less-sensitive weight profile.

## 4 EXPERIMENTS

To test the performance of the proposed MPPD-TV-$\ell_1$ framework in improving the robustness of SNNs, we basically follow the evaluation baseline of (Ding et al., 2022; 2024a) to conduct image classification experiments. The code is available at https://github.com/laizhr/MPPD-TV.

### 4.1 EXPERIMENTAL SETUP

**In the training stage**, VGG11 (Simonyan & Zisserman, 2015) and WRN16 (WideResNet-16-4,(Zagoruyko & Komodakis, 2017)) with Dynamic LIF (DLIF, (Ding et al., 2024a)) neurons are used as backbones of SNNs, while CIFAR 10, CIFAR 100 (Krizhevsky, 2009), and Tiny ImageNet (Le & Yang, 2015) are used as data sets. CIFAR 10 and CIFAR 100 have 60000 $32 \times 32$ images, categorized into 10 and 100 classes, respectively. Tiny ImageNet is a large-scale data set with 500 $64 \times 64$ downsized images for each of the 200 classes. In the training procedure, the time-step for SNN to infer forward is set to 8. Gaussian noise and adversarial noise (AT, Wong et al. 2020) are used as perturbations to construct training samples. In addition, the adversarial noise together with the regularizer of (Ding et al., 2022) is also used (AT+Reg). Perturbation strengths are set to $\zeta = 10/255$ for Gaussian noise, $\zeta = 6/255$ for AT, and $\zeta = 7/255$ for AT+Reg. The SGD optimizer is used with a starting learning rate of 0.01, then the learning rate is reduced to zero via cosine annealing.

**In the test stage**, the FGSM (Goodfellow et al., 2015), C&W (Carlini & Wagner, 2017), PGD (Madry et al., 2018), Auto-PGD (APGD), and AutoAttack (Croce & Hein, 2020) schemes are used to construct adversarial test samples, with attack intensity uniformly set to $\zeta = 8/255$. The number of steps for PGD ranges from 7 to 40, while the 10-step APGD based on the cross-entropy (CE) loss and the difference-of-logits-ratio (DLR) is used. All these settings including $\zeta$ strictly follow those of (Ding et al., 2024a) to make fair comparisons.

Eight state-of-the-art methods are taken into comparisons: SNN-BP (Sharmin et al., 2020), HIRE-SNN (Kundu et al., 2021), SNN-RAT (Ding et al., 2022), FEEL (Xu et al., 2024), SR (Liu et al., 2024), ANN-PGD-AT (Madry et al., 2018), ANN-RiFT (Zhu et al., 2023), and MPPD-TV-$\ell_2$ (Ding

---
[1]https://pytorch.org/

et al., 2024a). SNN-BP is a deep SNN with inherent adversarial robustness based on discrete input encoding and non-linear activations. HIRE-SNN is an energy-efficient deep SNN that can harness the inherent robustness. SNN-RAT is an SNN with regularized adversarial training that can enhance robustness. FEEL is an SNN with frequency encoding and evolutionary leak factor. SR is an SNN with sparsity regularization of gradients. MPPD-TV-$\ell_2$ is originally a kind of robust stable SNNs, which is found to be a TV-$\ell_2$ framework in the context of this paper. Hence it is denoted by MPPD-TV-$\ell_2$ to be comparable to the proposed MPPD-TV-$\ell_1$ framework. Besides, standard SNNs without MPPD (Non-MPPD) are also taken into comparisons as ablation studies. The default settings of these competitors are used in the experiments, where the regularization strength is set to $\alpha = 1$ for both MPPD-TV-$\ell_2$ and MPPD-TV-$\ell_1$.

Table 1: Classification accuracies (%) of different methods on CIFAR 10 and CIFAR 100.

| Perturbation | Model | Clean | APGD$_{CE}^{10}$ | APGD$_{DLR}^{10}$ | FGSM | PGD$^7$ | PGD$^{10}$ | PGD$^{20}$ | PGD$^{40}$ | CW | AutoAttack |
|---|---|---|---|---|---|---|---|---|---|---|---|
| | | | | | CIFAR 10 | | | | | | |
| | SNN-BP,VGG5 | 89.3 | - | - | 15.0 | 3.8 | - | - | - | - | - |
| | HIRE-SNN,VGG5 | 87.9 | - | - | 35.5 | 5.3 | - | - | - | - | - |
| | SNN-RAT,VGG11 | 90.74 | - | - | 45.23 | 21.16 | - | - | - | - | - |
| | FEEL+AT,VGG11 | 87.850 | 21.960 | 32.620 | 41.920 | 30.060 | 28.220 | 19.970 | 19.540 | 53.580 | 1.630 |
| | SR,VGG11 | 88.980 | 27.340 | 28.400 | 42.810 | 30.360 | 30.230 | 31.040 | 31.150 | 59.550 | 22.870 |
| | ANN-PGD-AT,VGG11 | 78.630 | 33.240 | 35.070 | 44.040 | 35.840 | 34.900 | 34.440 | 34.360 | 56.650 | 20.420 |
| | ANN-RiFT,VGG11 | 80.980 | 30.100 | 32.970 | 41.050 | 35.390 | 35.300 | 35.310 | 35.100 | 55.250 | 22.310 |
| | ANN-PGD-AT,WRN16 | 79.870 | 30.640 | 27.460 | 34.240 | 34.120 | 34.960 | 34.790 | 34.310 | 60.930 | 19.830 |
| | ANN-RiFT,WRN16 | 81.260 | 31.080 | 24.830 | 36.850 | 36.800 | 36.810 | 36.660 | 35.870 | 61.140 | 20.010 |
| Gaussian | Non-MPPD,VGG11 | 91.410 | 0.130 | 0.220 | 13.100 | 0.220 | 0.160 | 0.110 | 0.110 | 10.010 | 0.000 |
| | MPPD-TV-$\ell_2$,VGG11 | 90.990 | 0.130 | 0.160 | 15.160 | 0.230 | 0.110 | 0.060 | 0.060 | 10.410 | 0.000 |
| | **MPPD-TV-$\ell_1$,VGG11** | **92.230** | **0.340** | **0.400** | **20.250** | **1.140** | **0.620** | **0.410** | **0.370** | **13.030** | **0.290** |
| | Non-MPPD,WRN16 | 91.050 | 0.020 | 0.040 | 11.780 | 0.080 | 0.020 | 0.020 | 0.020 | 7.500 | 0.000 |
| | MPPD-TV-$\ell_2$,WRN16 | 90.520 | 0.030 | 0.030 | 12.540 | 0.100 | 0.040 | 0.030 | 0.040 | 8.840 | 0.010 |
| | **MPPD-TV-$\ell_1$,WRN16** | **92.390** | **0.060** | **0.040** | **15.350** | **0.420** | **0.270** | **0.180** | **0.150** | **10.520** | **0.010** |
| AT | Non-MPPD,VGG11 | 85.030 | 29.820 | 34.350 | 46.960 | 35.520 | 34.600 | 34.240 | 33.850 | 60.640 | 16.390 |
| | MPPD-TV-$\ell_2$,VGG11 | 85.170 | 27.780 | 35.300 | 46.510 | 34.510 | 33.200 | 32.260 | 32.470 | 63.050 | 19.75 |
| | **MPPD-TV-$\ell_1$,VGG11** | **86.110** | **36.590** | **45.260** | **51.890** | **42.840** | **41.560** | **41.150** | **40.850** | **66.680** | **23.040** |
| | Non-MPPD,WRN16 | 84.720 | 26.870 | 31.770 | 50.090 | 33.000 | 31.460 | 29.940 | 29.720 | 56.340 | 19.460 |
| | MPPD-TV-$\ell_2$,WRN16 | 84.380 | 30.270 | 34.150 | 49.950 | 35.650 | 34.030 | 33.430 | 32.660 | 59.110 | 21.340 |
| | **MPPD-TV-$\ell_1$,WRN16** | **86.340** | **32.320** | **37.440** | **52.500** | **39.040** | **37.900** | **36.740** | **36.290** | **63.870** | **22.920** |
| AT+Reg | Non-MPPD,VGG11 | 85.770 | 32.570 | 35.960 | 49.980 | 38.060 | 36.290 | 35.000 | 34.720 | 53.870 | 14.06 |
| | MPPD-TV-$\ell_2$,VGG11 | 84.910 | 33.620 | **39.490** | **54.520** | 39.030 | 36.570 | 34.530 | 33.270 | 54.340 | 19.070 |
| | **MPPD-TV-$\ell_1$,VGG11** | **86.390** | **35.160** | 38.630 | 50.970 | **40.730** | **39.070** | **38.060** | **37.670** | **62.400** | **23.530** |
| | Non-MPPD,WRN16 | 84.640 | 35.500 | 38.270 | 56.880 | 40.290 | 37.380 | 34.870 | 33.270 | 50.250 | 11.160 |
| | MPPD-TV-$\ell_2$,WRN16 | 84.220 | 33.530 | 37.430 | **58.320** | 39.100 | 35.800 | 32.700 | 31.310 | 53.570 | 13.69 |
| | **MPPD-TV-$\ell_1$,WRN16** | **85.400** | **36.680** | **39.580** | 57.440 | **41.490** | **38.260** | **35.900** | **34.780** | **60.580** | **18.010** |
| | | | | | CIFAR 100 | | | | | | |
| | SNN-BP,VGG11 | 64.4 | - | - | 15.5 | 6.3 | - | - | - | - | - |
| | HIRE-SNN,VGG11 | 65.6 | - | - | 16.4 | 2.9 | - | - | - | - | - |
| | SNN-RAT,VGG11 | 68.89 | - | - | 25.86 | 17.81 | - | - | - | - | - |
| | FEEL+AT,VGG11 | 66.530 | 15.440 | 15.380 | 16.680 | 5.560 | 5.310 | 8.020 | 7.930 | 14.880 | 0.810 |
| | SR,VGG11 | 67.930 | 11.530 | 11.740 | 19.690 | 13.160 | 13.180 | 13.750 | 13.740 | 25.350 | 10.540 |
| | ANN-PGD-AT,VGG11 | 47.050 | 15.120 | 15.630 | 20.340 | 16.350 | 15.850 | 15.750 | 15.600 | 34.760 | 7.470 |
| | ANN-RiFT,VGG11 | 48.880 | 15.710 | 16.550 | 21.840 | 21.730 | 21.720 | 21.710 | 21.710 | 34.160 | 8.320 |
| | ANN-PGD-AT,WRN16 | 55.040 | 15.270 | 19.070 | 21.420 | 18.970 | 18.890 | 18.110 | 18.590 | 34.250 | 4.290 |
| | ANN-RiFT,WRN16 | 52.480 | 14.620 | 18.560 | 18.560 | 18.540 | 18.540 | 18.520 | 18.360 | 35.010 | 7.870 |
| Gaussian | Non-MPPD,VGG11 | 68.770 | 0.540 | 1.120 | 8.330 | 0.980 | 0.710 | 0.690 | 0.570 | 11.030 | 0.080 |
| | MPPD-TV-$\ell_2$,VGG11 | 68.900 | 0.390 | 1.080 | 8.470 | 0.690 | 0.540 | 0.470 | 0.350 | 13.340 | 0.090 |
| | **MPPD-TV-$\ell_1$,VGG11** | **69.410** | **0.820** | **1.520** | 8.680 | **1.390** | **1.150** | **1.070** | **0.960** | 12.640 | **0.250** |
| | Non-MPPD,WRN16 | 66.260 | 0.290 | 0.700 | 8.700 | 0.450 | 0.330 | 0.290 | 0.210 | 9.680 | 0.000 |
| | MPPD-TV-$\ell_2$,WRN16 | 65.990 | 0.190 | 0.810 | **9.070** | 0.410 | 0.290 | 0.140 | 0.090 | **13.130** | **0.110** |
| | **MPPD-TV-$\ell_1$,WRN16** | **67.770** | 0.350 | **1.010** | 8.210 | **0.560** | **0.430** | **0.350** | **0.360** | 12.570 | 0.050 |
| AT | Non-MPPD,VGG11 | 56.370 | 16.460 | 19.400 | 25.260 | 19.950 | 19.300 | 19.140 | 19.010 | 34.170 | 8.560 |
| | MPPD-TV-$\ell_2$,VGG11 | 57.820 | 12.920 | 16.690 | 24.550 | 16.440 | 15.600 | 15.180 | 14.960 | 34.650 | 8.450 |
| | **MPPD-TV-$\ell_1$,VGG11** | **58.410** | **16.600** | **19.670** | 25.720 | **20.570** | **19.970** | **19.710** | **19.440** | 35.210 | **11.590** |
| | Non-MPPD,WRN16 | 55.580 | 16.530 | 20.490 | 29.580 | 20.110 | 18.980 | 18.080 | 17.950 | 37.810 | 6.470 |
| | MPPD-TV-$\ell_2$,WRN16 | 54.720 | 13.570 | 17.620 | 25.790 | 16.850 | 16.050 | 15.410 | 15.030 | 38.630 | 8.750 |
| | **MPPD-TV-$\ell_1$,WRN16** | **56.060** | **16.630** | 19.830 | 27.400 | 20.000 | **19.270** | **18.480** | 18.420 | 39.370 | **9.140** |
| AT+Reg | Non-MPPD,VGG11 | 62.190 | 21.550 | 23.440 | 34.370 | 24.680 | 22.650 | 20.850 | 20.060 | 35.820 | 6.260 |
| | MPPD-TV-$\ell_2$,VGG11 | 61.980 | 19.480 | **24.220** | **35.940** | 23.010 | 20.380 | 17.940 | 16.650 | 36.640 | 5.680 |
| | **MPPD-TV-$\ell_1$,VGG11** | **62.710** | **21.740** | 23.700 | 34.390 | **25.360** | **23.650** | **21.520** | **20.620** | **39.710** | **10.800** |
| | Non-MPPD,WRN16 | 53.740 | 15.730 | 19.110 | 28.710 | 18.220 | 16.960 | 15.980 | 15.290 | 31.880 | 4.330 |
| | MPPD-TV-$\ell_2$,WRN16 | 54.010 | 15.510 | **21.550** | **33.000** | 19.210 | 17.090 | 15.270 | 14.390 | 33.870 | 5.350 |
| | **MPPD-TV-$\ell_1$,WRN16** | **54.140** | **17.910** | 20.790 | 29.770 | **20.550** | **19.100** | **17.870** | **17.300** | **35.450** | **8.490** |

Table 2: Classification accuracies (%) of different methods on Tiny ImageNet.

| Perturbation | Model | Clean | APGD$_{CE}^{10}$ | APGD$_{DLR}^{10}$ | FGSM | PGD$^7$ | PGD$^{10}$ | PGD$^{20}$ | PGD$^{40}$ | CW | AutoAttack |
|---|---|---|---|---|---|---|---|---|---|---|---|
| | FEEL+AT,VGG11 | 55.230 | 8.380 | 8.930 | 15.860 | 10.670 | 10.270 | 10.150 | 10.090 | 1.720 | 0.570 |
| | SR,VGG11 | 56.010 | 8.510 | 8.780 | 16.940 | 11.960 | 11.580 | 11.370 | 11.180 | 2.450 | 2.880 |
| | ANN-PGD-AT,VGG11 | 23.330 | 1.290 | 1.980 | 5.200 | 2.380 | 2.150 | 2.070 | 1.600 | 0.760 | 0.470 |
| | ANN-RiFT,VGG11 | 24.510 | 1.400 | 2.460 | 6.780 | 2.270 | 2.170 | 1.830 | 1.610 | 1.220 | 0.620 |
| | ANN-PGD-AT,WRN16 | 15.040 | 0.670 | 0.900 | 3.510 | 1.470 | 1.280 | 1.010 | 0.690 | 0.620 | 0.190 |
| | ANN-RiFT,WRN16 | 14.670 | 1.030 | 1.670 | 3.270 | 1.690 | 1.440 | 1.110 | 0.860 | 0.510 | 0.070 |
| Gaussian | Non-MPPD,VGG11 | 54.280 | 2.020 | 2.260 | 9.870 | 3.380 | 3.040 | 2.910 | 2.950 | 0.820 | 10.590 |
| | MPPD-TV-$\ell_2$,VGG11 | 55.470 | 1.980 | 2.380 | 10.110 | 3.420 | 3.290 | 3.430 | 3.290 | 1.990 | 11.460 |
| | **MPPD-TV-$\ell_1$,VGG11** | **56.530** | **2.380** | **2.770** | **10.340** | **3.770** | **3.660** | **3.580** | **3.470** | **2.500** | **12.310** |
| | Non-MPPD,WRN16 | 43.110 | 1.970 | 1.630 | 6.380 | 2.350 | 2.270 | 2.190 | 2.070 | 0.740 | 1.070 |
| | MPPD-TV-$\ell_2$,WRN16 | 44.290 | 2.070 | 1.980 | 6.540 | 2.520 | 2.410 | 2.370 | 2.240 | 2.080 | 1.230 |
| | **MPPD-TV-$\ell_1$,WRN16** | **46.750** | **2.820** | **2.780** | **7.050** | **3.720** | **3.710** | **3.590** | **3.680** | **2.460** | **1.380** |
| AT | Non-MPPD,VGG11 | 47.880 | 8.750 | 9.840 | 18.310 | 13.940 | 12.830 | 12.570 | 11.830 | 0.980 | 3.740 |
| | MPPD-TV-$\ell_2$,VGG11 | 48.380 | 8.350 | 9.460 | 19.450 | 13.210 | 13.080 | 13.050 | 12.770 | 1.430 | 3.890 |
| | **MPPD-TV-$\ell_1$,VGG11** | **49.290** | **9.520** | **10.720** | **20.770** | **14.090** | **13.660** | **13.420** | **13.290** | **2.790** | **4.080** |
| | Non-MPPD,WRN16 | 33.620 | 3.710 | 4.030 | 12.250 | 6.950 | 6.310 | 6.110 | 6.040 | 1.400 | 2.070 |
| | MPPD-TV-$\ell_2$,WRN16 | 33.990 | 4.230 | 4.960 | 12.730 | 7.360 | 7.290 | 7.170 | 6.840 | 2.450 | 2.310 |
| | **MPPD-TV-$\ell_1$,WRN16** | **35.000** | **5.420** | **5.730** | **13.720** | **7.840** | **7.480** | **7.330** | **7.240** | **3.170** | **2.660** |
| AT+Reg | Non-MPPD,VGG11 | 49.390 | 8.850 | 9.740 | 22.090 | 12.820 | 12.640 | 12.340 | 12.290 | 1.390 | 5.570 |
| | MPPD-TV-$\ell_2$,VGG11 | 50.720 | 9.740 | 10.080 | 23.080 | 13.970 | 13.610 | 13.480 | 13.250 | 1.270 | 6.290 |
| | **MPPD-TV-$\ell_1$,VGG11** | **52.990** | **10.440** | **11.050** | **23.440** | **14.290** | **13.980** | **13.520** | **13.430** | **2.990** | **6.720** |
| | Non-MPPD,WRN16 | 28.660 | 6.310 | 5.920 | 12.470 | 7.770 | 7.540 | 7.390 | 7.280 | 1.190 | 3.750 |
| | MPPD-TV-$\ell_2$,WRN16 | 29.090 | 6.720 | 6.180 | 12.350 | 8.280 | 7.940 | 7.770 | 7.490 | 2.580 | 3.950 |
| | **MPPD-TV-$\ell_1$,WRN16** | **31.240** | **7.060** | **6.740** | **13.260** | **9.720** | **9.500** | **9.450** | **9.280** | **3.110** | **4.210** |

Table 3: Classification accuracies (%) of MPPD-TV-$\ell_1$ with different regularization strengths.

| Model | Clean | APGD$_{CE}^{10}$ | APGD$_{DLR}^{10}$ | FGSM | PGD$^7$ | PGD$^{10}$ | PGD$^{20}$ | PGD$^{40}$ | CW | AutoAttack |
|---|---|---|---|---|---|---|---|---|---|---|
| AT,$\alpha = 0.0$ | 82.990 | 26.370 | 29.940 | 40.360 | 30.990 | 29.860 | 29.460 | 29.540 | 51.910 | 18.870 |
| AT,$\alpha = 0.5$ | 83.250 | 28.100 | 31.420 | 41.400 | 32.470 | 31.470 | 31.380 | 30.860 | 54.800 | 20.800 |
| AT,$\alpha = 1.0$ | 83.640 | 29.250 | 31.940 | 42.230 | 33.350 | 32.620 | 32.150 | 31.830 | 55.640 | 22.270 |
| AT,$\alpha = 2.0$ | 82.860 | 30.090 | 32.840 | 42.800 | 34.140 | 33.450 | 32.960 | 32.820 | 57.660 | 23.190 |
| AT,$\alpha = 2.5$ | 83.750 | 30.640 | 33.560 | 43.580 | 34.690 | 33.910 | 33.400 | 33.280 | 57.690 | 22.760 |
| AT,$\alpha = 3.0$ | 83.550 | 30.430 | 33.240 | 43.330 | 34.540 | 33.420 | 33.120 | 32.780 | 60.070 | 25.020 |
| AT,$\alpha = 3.5$ | 84.010 | 30.460 | 33.910 | 43.630 | 34.680 | 33.830 | 33.360 | 33.130 | 56.670 | 13.590 |
| AT,$\alpha = 4.0$ | 83.470 | 30.850 | 33.470 | 43.490 | 34.640 | 33.610 | 33.070 | 33.140 | 58.050 | 22.340 |

## 4.2 EXPERIMENTAL RESULTS

Image classification results of different methods are provided in Tables 1 and 2. MPPD-TV-$\ell_1$ outperforms other competitors in most cases for both VGG11 and WRN16 architectures on all of CIFAR 10, CIFAR 100, and Tiny ImageNet data sets. Taking the VGG11 architecture with the AT training scheme on CIFAR 10 as an example, MPPD-TV-$\ell_1$ achieves classification accuracies of $(45.260\%, 51.890\%, 42.840\%)$, compared with $(34.350\%, 46.960\%, 35.520\%)$ of Non-MPPD and $(35.300\%, 46.510\%, 34.510\%)$ of MPPD-TV-$\ell_2$ for the APGD$_{DLR}^{10}$, FGSM, and PGD$^7$ attacks, respectively. Besides, MPPD-TV-$\ell_1$ outperforms MPPD-TV-$\ell_2$ and Non-MPPD on both clean and perturbed data. This indicates that MPPD-TV-$\ell_1$ really improves robustness not just against adversarial perturbations, but also against other types of detrimental noise. Note that AT+Reg is already a heavy double penalization for non-robust dynamics. The fact that MPPD-TV-$\ell_1$ shows little further improvement under AT+Reg suggests that MPPD-TV-$\ell_1$ is implicitly achieving the desired robust regularization effect that the explicit Reg treatment of (Ding et al., 2022) is designed for. In the more common and important training scenario AT, MPPD-TV-$\ell_1$ consistently shows better performance, which proves its practical necessity and advantage as a standalone, effective robust training method. These results indicate that MPPD-TV-$\ell_1$ is effective in suppressing adversarial perturbations. The runtimes of different methods with VGG11 and AT on the three data sets are provided in Table A1, which indicates that MPPD-TV-$\ell_1$ runs the fastest among the competitors. The gradient magnitudes for different methods with WRN16 architecture and AT training scheme on Tiny ImageNet are provided in Figure A1, which show that MPPD-TV-$\ell_1$ converges quickly to a low gradient magnitude level around the 400-th iteration, and maintains the lowest gradient magnitude among the competitors.

Moreover, MPPD-TV-$\ell_1$ achieves more robust performance than MPPD-TV-$\ell_2$, especially for the PGD attacks. For instance, when training both VGG11 and WRN16 architectures with Gaussian noise on CIFAR 100, MPPD-TV-$\ell_1$ achieves significantly higher classification accuracies than MPPD-TV-$\ell_2$ on all the PGD attacks. Specifically, the accuracies of MPPD-TV-$\ell_1$ with VGG11 on $PGD^7$, $PGD^{10}$, $PGD^{20}$, and $PGD^{40}$ are 1.390%, 1.150%, 1.070%, and 0.960%, respectively, which are significantly higher than those of MPPD-TV-$\ell_2$: 0.690%, 0.540%, 0.470%, and 0.350%. Moreover, as the number of iterative steps increases for the PGD attack, the gap between MPPD-TV-$\ell_1$ and MPPD-TV-$\ell_2$ also increases. It indicates that MPPD-TV-$\ell_1$ is more advantageous when the perturbations get more adversarial.

### 4.3 REGULARIZATION STRENGTH $\alpha$

To investigate the impact of regularization strength $\alpha$, we use the VGG5 model to conduct experiments on CIFAR 10, shown in Table 3. The values of $\alpha$ are set to $0.0 \sim 4.0$, respectively. Results show that MPPD-TV-$\ell_1$ achieves higher accuracies with $\alpha > 0$ than those with $\alpha = 0$ against adversarial attacks, which indicates that MPPD-TV-$\ell_1$ is effective in extracting and suppressing such adversarial perturbations. As $\alpha$ varies, the accuracies of MPPD-TV-$\ell_1$ reach their peaks around $\alpha = 2.5 \sim 3.0$.

Next, we evaluate the adversarial robustness of MPPD-TV-$\ell_1$ with a VGG11 architecture pre-trained on CIFAR 10 by subjecting it to $PGD^{10}$ attacks with gradually increasing intensity, then plot the resulting accuracy curves in Figures 1a and 1b. Specifically, we increase the attack intensity $\zeta$ from $10/255$ to $100/255$ by increments of $10/255$. Results indicate that the MPPD-TV-$\ell_1$ curves ($\alpha = 1$) decrease more gradually than the Non-MPPD curves ($\alpha = 0$) as the intensity increases, especially with AT training samples. We also calculate the actual TV values for MPPD-TV-$\ell_1$ and Non-MPPD, shown in Figures 1c and 1d. Results indicate that MPPD-TV-$\ell_1$ ($\alpha = 1$) indeed produces less TV than Non-MPPD ($\alpha = 0$), which accords with the design intention of MPPD-TV-$\ell_1$.

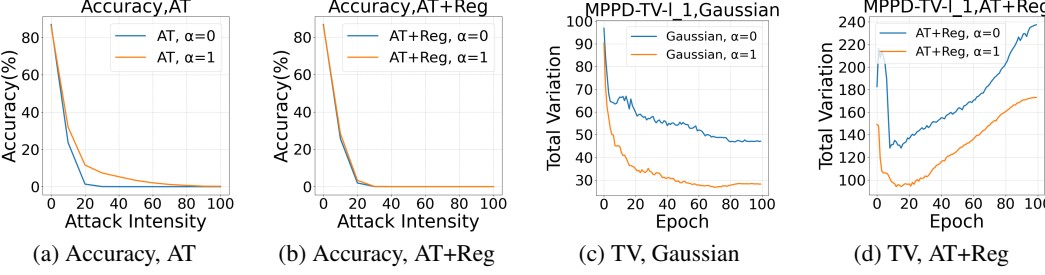

| (a) Accuracy, AT | (b) Accuracy, AT+Reg | (c) TV, Gaussian | (d) TV, AT+Reg |

Figure 1: Accuracies and actual TV values of MPPD-TV-$\ell_1$ ($\alpha = 1$) and Non-MPPD ($\alpha = 0$).

## 5 CONCLUSION

Membrane potential perturbation dynamic (MPPD) is a new method to capture and suppress adversarial perturbations for spiking neural networks (SNN). However, it discards the neuronal reset part without reliable theoretical foundation. To fix this problem, we formulate that MPPD is total variation (TV) and its regularization scheme is essentially a TV-$\ell_2$ model (MPPD-TV-$\ell_2$). Based on this insight, we propose the MPPD-TV-$\ell_1$ model to further improve the robustness of SNNs. Because the $L^1$ function space is larger than the $L^2$ function space with finite measures, MPPD-TV-$\ell_1$ facilitates broader classes of functions to be membrane potentials, thus expands its applicability and flexibility. Moreover, MPPD-TV-$\ell_1$ can exploit the coarea formula while MPPD-TV-$\ell_2$ cannot, hence the former has better performance than MPPD-TV-$\ell_2$ in robust signal reconstruction against adversarial perturbations, which better fits the architectures of SNNs. The only fundamental requirement of the proposed theory is that the perturbation is a measurable function of the node index and the time-step of an SNN, otherwise this perturbation cannot be captured to yield significant TV.

Experimental results show that the MPPD-TV-$\ell_1$ framework achieves better performance than other state-of-the-art methods in most test scenarios, and shows better robustness in complicated environments with adversarial perturbations and signal distortions. In summary, we establish a theoretically-sound TV formulation for MPPD, which provides a new insight into the essence of perturbation characterization for SNNs. Our methodology is applicable to most SNN architectures where a TV term is used to stabilize layer-wise internal state. Future works may lie in applying the above theory to improve robustness of neuromorphic computing systems in safety-critical applications, such as autonomous driving and industrial control.

ACKNOWLEDGMENTS

This work is supported in part by the National Natural Science Foundation of China (grant numbers: 62541606, 62176103, 62206110, and 62276114), in part by The Major Key Project of PCL (No. PCL2025A02 and No. PCL2024A04), and in part by the National Research Foundation, Singapore and Infocomm Media Development Authority under its Trust Tech Funding Initiative. Any opinions, findings and conclusions or recommendations expressed in this material are those of the author(s) and do not reflect the views of National Research Foundation, Singapore and Infocomm Media Development Authority.

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

# A  APPENDIX

## A.1  PROOF OF THEOREM 1

*Proof.* **Part (1):** We first verify that the following local variation is well-defined:

$$\nabla_{(i,t)}v(i,t,x) := v(i,t,x) - v(i,t,x + \delta(i,t)). \tag{27}$$

Let $x, \delta \in \mathbb{R}^d$ and $(i,t) \in \Theta$, where the domain $\Theta$ can be $[0,N] \times [0,T]$ for the continuous setting, $[0:N] \times [0:T]$ for the discrete setting, or $[0:N] \times [0,T]$ or $[0,N] \times [0:T]$ for the mixed setting. Denote the $\sigma$-algebras of $\Theta$, $\mathbb{R}^d$, and $\mathbb{R}$ by $\mathscr{F}$, $\mathscr{G}$, and $\mathscr{H}$, respectively. $\mathscr{F}$ can take the product $\sigma$-algebra w.r.t. its two arguments $i$ and $t$. For either argument, the power set or the Lebesgue $\sigma$-algebra can be used for the discrete or continuous setting, respectively. $\mathscr{G}$ and $\mathscr{H}$ take the $d$-dimensional and one-dimensional Lebesgue $\sigma$-algebras by default, respectively. The $\sigma$-algebra of $\Theta \times \mathbb{R}^d$ takes the product $\sigma$-algebra $\mathscr{F} \times \mathscr{G}$.

As a necessary condition, $v : \Theta \times \mathbb{R}^d \mapsto \mathbb{R}$ should be a measurable function of $(i,t,x)$ for an eligible SNN, otherwise this SNN cannot process the input information. Because $\delta : \Theta \mapsto \mathbb{R}^d$ is measurable, given any set $\mathcal{F} \in \mathscr{F}$, we have $\delta(\mathcal{F}) \in \mathscr{G}$. Consider $x$ as a fixed point in $\mathbb{R}^d$, then $(x + \delta(\mathcal{F}))$ is a translation of $\delta(\mathcal{F})$. According to the property of Lebesgue $\sigma$-algebra, $(x + \delta(\mathcal{F})) \in \mathscr{G}$. Therefore, $\mathcal{F} \times (x + \delta(\mathcal{F})) \in \mathscr{F} \times \mathscr{G}$ and $v(\mathcal{F} \times (x + \delta(\mathcal{F}))) \in \mathscr{H}$ from the measurability of $v$. Hence the forward mapping of $v$ is well-defined.

Conversely, given any set $\mathcal{H} \in \mathscr{H}$, the preimage $v^{-1}(\mathcal{H}) = \mathcal{F} \times \mathcal{G} \in \mathscr{F} \times \mathscr{G}$ from the measurability of $v$. Again from the property of Lebesgue $\sigma$-algebra, the translation $(\mathcal{G} - x) \in \mathscr{G}$. Then from the measurability of $\delta$, the preimage $\delta^{-1}(\mathcal{G} - x) \in \mathscr{F}$. Denote the intersection of $\mathcal{F}$ and $\delta^{-1}(\mathcal{G} - x)$ by $\mathcal{F}' := \mathcal{F} \cap \delta^{-1}(\mathcal{G} - x)$. Since the $\sigma$-algebra $\mathscr{F}$ is closed under intersection, $\mathcal{F}' \in \mathscr{F}$. By fixing $x$ as a constant function of $(i,t)$, we consider $v$ as a composed function: $v \circ (x + \delta) : \Theta \mapsto \mathbb{R}$. Then the above deduction indicates that the preimage $(v \circ (x + \delta))^{-1}(\mathcal{H}) = \mathcal{F}' \in \mathscr{F}$. Hence $v \circ (x + \delta)$ is also a measurable function of $(i,t)$ and the inverse mapping $(v \circ (x + \delta))^{-1}$ is well-defined.

Summarizing the above deductions, we verify that $v(i,t,x+\delta(i,t))$ is a measurable function of $(i,t)$ for any fixed $x$. Since $\nabla_{(i,t)}v(i,t,x)$ in (27) is a subtraction between two measurable functions, it is also a measurable function of $(i,t)$ for any fixed $x$. Hence $\nabla_{(i,t)}v(i,t,x)$ is well-defined and can be calculated in practice.

**Part (2):** Next, we need to verify that $\int_{\mathcal{J}(i)} \nabla_{(j,t)}s(j,t,x)\,\mathrm{d}w(i,j(i))$ is well-defined. The spike function can be rewritten as:

$$s(j,t,x) = H(v(j,t,x) - u_{th}). \tag{28}$$

Since $(v(j,t,x) - u_{th})$ and the Heaviside function are both measurable functions, their composite $s(j,t,x)$ is also a measurable function. Following similar deductions to Part (1), the local variation $\nabla_{(j,t)}s(j,t,x)$ is also a well-defined measurable function. With a fixed $i$, the weight function $w(i,j(i))$ is naturally a measure on $\mathcal{J}(i)$. With fixed $t$ and $x$, $\nabla_{(j,t)}s(j,t,x)$ is also a measurable function restricted on $\mathcal{J}(i)$. Hence $\int_{\mathcal{J}(i)} \nabla_{(j,t)}s(j,t,x)\,\mathrm{d}w(i,j(i))$ is a well-defined Lebesgue integral. Moreover, since $|\nabla_{(j,t)}s(j,t,x)| \leqslant 1$ from the definition of the Heaviside function, $\int_{\mathcal{J}(i)} \nabla_{(j,t)}s(j,t,x)\,\mathrm{d}w(i,j(i))$ is also a finite integral with a finite measure $w(i,j(i))$. This is crucial for the dominated TV property of Theorem 4 that controls the overall stability of an SNN.

Again by similar deductions to Part (1), $\nabla_{(i,t)}v(i,t-1,x)$ is also a well-defined measurable function. Hence (19) holds in a well-defined measurable sense. As for (20), we can use either counting measure or Lebesgue measure for the discrete or continuous setting of $i$ and $t$, respectively. Then both sides of (20) are well-defined Lebesgue integrals, forming a TV-$\ell_2$ term. Moreover, this MPPD-TV-$\ell_2$ term is finite in general situations, as stated in Theorem 4.

$\square$

## A.2  PROOF OF THEOREM 2

*Proof.* To simplify notations, we can fix and omit the input variable $x$ in the rest of the appendices if not specified. We use the notations in Appendix A.1. Since $v$ is measurable, given any set $\mathcal{H} \in \mathscr{H}$, $(v|_x)^{-1}(\mathcal{H}) \in \mathscr{F}$. On the other hand, the interval type $[\psi, \psi + \Delta\psi) \in \mathscr{H}$ from the definition

of Lebesgue $\sigma$-algebra. The main technique to calculate the TV-$\ell_1$ term in (22) is to partition this Lebesgue integral w.r.t. the values of $v$ along with $(-\infty, \infty)$. To do this, we observe that rational numbers are dense in $(-\infty, \infty)$. Since rational numbers are countable, we can construct a countable set of $M \in \mathbb{N}^+ \cup \{+\infty\}$ intervals with positive Lebesgue measure (i.e., positive length), as follows.

$$\{B_m := [a_m, b_m)\}_{m=1}^M \quad \text{s.t.} \quad a_m < b_m \leqslant a_{m+1}, \quad m = 1, 2, \cdots, M.$$
$$v(i,t) \text{ is Lipschitz continuous on } \mathcal{F}_m := \{(i,t) \in \Theta : a_m \leqslant v(i,t) < b_m\}. \tag{29}$$

Each interval $[a_m, b_m)$ contains at least one rational number, and all these intervals are mutually disjoint: $B_m \cap B_o = \emptyset$ for any $m \neq o$. Hence $\cup_{m=1}^M B_m$ covers all the Lipschitz continuous intervals of the range of $v$. We only need to consider preimage sets $\{\mathcal{F}_m\}_{m=1}^M$ where $v$ is Lipschitz continuous because the corresponding Lebesgue integrals are positive only on these sets. Specifically,

$$\int_\Theta |\nabla_{(i,t)} v(i,t)|\, d\mu = \int_{\Theta \backslash (\cup_{m=1}^M \mathcal{F}_m)} |\nabla_{(i,t)} v(i,t)|\, d\mu + \int_{\cup_{m=1}^M \mathcal{F}_m} |\nabla_{(i,t)} v(i,t)|\, d\mu, \tag{30}$$

where $\Theta \backslash (\cup_{m=1}^M \mathcal{F}_m)$ corresponds to $\mathbb{R} \backslash (\cup_{m=1}^M B_m)$ where $v$ is discontinuous w.r.t. $(i,t)$ almost everywhere (a.e.). Hence $\int_{\Theta \backslash (\cup_{m=1}^M \mathcal{F}_m)} |\nabla_{(i,t)} v(i,t)|\, d\mu = 0$ based on the definition of Lebesgue integral, which means that it has zero volume. Then we just need to calculate $\int_{\cup_{m=1}^M \mathcal{F}_m} |\nabla_{(i,t)} v(i,t)|\, d\mu$. We break this down into the discrete and the continuous settings.

**Part (1):** For the **discrete setting**, direct calculation yields:

$$\int_{\mathcal{F}_m} |\nabla_{(i,t)} v(i,t)|\, d\mu = \varphi(\mathcal{F}_m) \cdot (b_m - a_m), \quad \forall m. \tag{31}$$

Since $\varphi(\mathcal{F}_m)$ remains unchanged in the interval $v \in [a_m, b_m)$ due to Lipschitz continuity, we have $\mathcal{F}_m = \{(i,t) \in \Theta : v(i,t) = a_m\}$. By letting $\psi_m = a_m$ and $\Delta\psi_m = b_m - a_m$, (31) can be reformulated as

$$\varphi(\mathcal{F}_m) \cdot (b_m - a_m) = \varphi(\{(i,t) \in \Theta : v(i,t) = \psi_m\}) \cdot \Delta\psi_m = \int_{\{(i,t) \in \Theta : v(i,t) = \psi_m\}} d\varphi\, d\psi, \quad \forall m. \tag{32}$$

From the $\sigma$-additivity of Lebesgue integrals,

$$\int_{\cup_{m=1}^M \mathcal{F}_m} |\nabla_{(i,t)} v(i,t)|\, d\mu = \sum_{m=1}^M \int_{\mathcal{F}_m} |\nabla_{(i,t)} v(i,t)|\, d\mu$$
$$= \sum_{m=1}^M \int_{\{(i,t) \in \Theta : v(i,t) = \psi_m\}} d\varphi\, d\psi = \int_{\cup_{m=1}^M B_m} \int_{\{(i,t) \in \Theta : v(i,t) = \psi_m\}} d\varphi\, d\psi. \tag{33}$$

Adding the zero integral terms w.r.t. $\Theta \backslash (\cup_{m=1}^M \mathcal{F}_m)$ and $\mathbb{R} \backslash (\cup_{m=1}^M B_m)$ to both sides of (33) yields:

$$\int_\Theta |\nabla_{(i,t)} v(i,t)|\, d\mu = \int_{-\infty}^\infty \int_{\{(i,t) \in \Theta : v(i,t) = \psi\}} d\varphi\, d\psi, \tag{34}$$

which proves the coarea formula (22).

**Part (2):** For the **continuous setting**, we can use the existing calculation for each Lipschitz continuous interval (Federer, 1959):

$$\int_{\mathcal{F}_m} |\nabla_{(i,t)} v(i,t)|\, d\mu = \int_{a_m}^{b_m} \int_{\{(i,t) \in \Theta : v(i,t) = \psi\}} d\varphi\, d\psi, \quad \forall m. \tag{35}$$

Similar to the deductions in Part (1), we exploit the $\sigma$-additivity of Lebesgue integrals and add the zero integral terms to obtain:

$$\int_{\cup_{m=1}^M \mathcal{F}_m} |\nabla_{(i,t)} v(i,t)|\, d\mu = \sum_{m=1}^M \int_{\mathcal{F}_m} |\nabla_{(i,t)} v(i,t)|\, d\mu$$

$$= \sum_{m=1}^{M} \int_{a_m}^{b_m} \int_{\{(i,t)\in\Theta:v(i,t)=\psi\}} \mathrm{d}\varphi\,\mathrm{d}\psi = \int_{\cup_{m=1}^{M} B_m} \int_{\{(i,t)\in\Theta:v(i,t)=\psi\}} \mathrm{d}\varphi\,\mathrm{d}\psi,$$

$$\int_{\Theta} |\nabla_{(i,t)}v(i,t)|\,\mathrm{d}\mu = \int_{-\infty}^{\infty} \int_{\{(i,t)\in\Theta:v(i,t)=\psi\}} \mathrm{d}\varphi\,\mathrm{d}\psi. \tag{36}$$

For the mixed setting (with $i$ discrete and $t$ continuous, or $t$ discrete and $i$ continuous), the proof is similar to the above, which is omitted here.

$\square$

## A.3 PROOF OF THEOREM 3

*Proof.* The proof is basically the same as that of Theorem 1 in Appendix A.1 except that the integrated function takes the absolute form $|\cdot|$ instead of the squared form $|\cdot|^2$, thus we need not repeat it again. Moreover, the MPPD-TV-$\ell_1$ term in (23) is finite according to Theorem 4, which can be calculated and quantified in practice.

Next, we verify that the function space $L^1(\Theta) \supsetneq L^2(\Theta)$ when $\mu(\Theta) < \infty$, so that MPPD-TV-$\ell_1$ allows for broader classes of functions than MPPD-TV-$\ell_2$:

$$\int_{\Theta} |\nabla_{(i,t)}v(i,t)|\,\mathrm{d}\mu$$

$$= \int_{\Theta\cap\{(i,t):|\nabla_{(i,t)}v(i,t)|>1\}} |\nabla_{(i,t)}v(i,t)|\,\mathrm{d}\mu + \int_{\Theta\cap\{(i,t):0\leqslant|\nabla_{(i,t)}v(i,t)|\leqslant 1\}} |\nabla_{(i,t)}v(i,t)|\,\mathrm{d}\mu$$

$$\leqslant \int_{\Theta\cap\{(i,t):|\nabla_{(i,t)}v(i,t)|>1\}} |\nabla_{(i,t)}v(i,t)|^2\,\mathrm{d}\mu + \int_{\Theta\cap\{(i,t):0\leqslant|\nabla_{(i,t)}v(i,t)|\leqslant 1\}} 1\cdot\mathrm{d}\mu \tag{37}$$

$$\leqslant \int_{\Theta} |\nabla_{(i,t)}v(i,t)|^2\,\mathrm{d}\mu + \mu(\Theta) \tag{38}$$

$$< \infty.$$

The inequality (37) holds because $|\nabla_{(i,t)}v(i,t)| \leqslant |\nabla_{(i,t)}v(i,t)|^2$ when $|\nabla_{(i,t)}v(i,t)| > 1$ in the first term, and $|\nabla_{(i,t)}v(i,t)| \leqslant 1$ in the second term. The inequality (38) holds due to the expansions of the integration intervals. Last, $\int_{\Theta} |\nabla_{(i,t)}v(i,t)|^2\,\mathrm{d}\mu < \infty$ implies $\int_{\Theta} |\nabla_{(i,t)}v(i,t)|\,\mathrm{d}\mu < \infty$. Hence $L^1(\Theta) \supseteq L^2(\Theta)$.

However, $L^1(\Theta) \neq L^2(\Theta)$. For a simple counterexample, we let $f(t) := t^{-\frac{1}{2}}$, $\Theta := [0,1]$ and use the Lebesgue measure. Then $\int_0^1 f(t)\,\mathrm{d}t < \infty$ but $\int_0^1 f^2(t)\,\mathrm{d}t = \infty$. Hence $f \in L^1(\Theta)$ but $f \notin L^2(\Theta)$. There are many other functions like this $f$.

$\square$

## A.4 PROOF OF THEOREM 4

*Proof.* **Part (1):** Without loss of generality, we define and use the following continuous version of (21) w.r.t. $k$:

$$\nabla_{(i,t)}v(i,t) = \int_0^{t-1} \lambda^k \int_{\mathcal{J}} \nabla_{(j,t)}s(j,t-k)\,\mathrm{d}w(i,j)\,\mathrm{d}k. \tag{39}$$

Then for the MPPD-TV-$\ell_1$ Case,

$$\int_1^{N^l} \int_1^T |\nabla_{(i,t)}v(i,t)|\,\mathrm{d}t\,\mathrm{d}i$$

$$= \int_1^T \left( \int_1^{N^l} |\nabla_{(i,t)}v(i,t)|\,\mathrm{d}i \right)\,\mathrm{d}t$$

$$= \int_1^T \left( \int_1^{N^l} \left| \int_0^{t-1} \lambda^k \int_{\mathcal{J}} \nabla_{(j,t)}s(j,t-k)w_l(i,j)\,\mathrm{d}j\,\mathrm{d}k \right|\,\mathrm{d}i \right)\,\mathrm{d}t \tag{40}$$

$$\leqslant \int_1^T \left( \int_0^{t-1} \lambda^k \int_1^{N^l} \int_{\mathcal{J}} \left| \nabla_{(j,t)} s(j, t-k) w_l(i,j) \right| \, \mathrm{d}j \, \mathrm{d}i \, \mathrm{d}k \right) \mathrm{d}t$$

$$= \int_1^T \left( \int_0^{t-1} \lambda^k \int_{\mathcal{J}} \left| \nabla_{(j,t)} s(j, t-k) \right| \underline{\left( \int_1^{N^l} |w_l(i,j)| \, \mathrm{d}i \right)} \, \mathrm{d}j \, \mathrm{d}k \right) \mathrm{d}t$$

$$\leqslant \int_1^T \left( \int_0^{t-1} \lambda^k \cdot \underline{\sup_{j \in \mathcal{J}} \left( \int_1^{N^l} |w_l(i,j)| \, \mathrm{d}i \right)} \int_{\mathcal{J}} \left| \nabla_{(j,t)} s(j, t-k) \right| \, \mathrm{d}j \, \mathrm{d}k \right) \mathrm{d}t$$

$$= \int_1^T \left( \int_0^{t-1} \lambda^k \underline{\|w_l\|_1} \int_{\mathcal{J}} \left| \nabla_{(j,t)} s(j, t-k) \right| \, \mathrm{d}j \, \mathrm{d}k \right) \mathrm{d}t$$

$$= \|w_l\|_1 \int_{\mathcal{J}} \left( \int_1^T \int_0^{t-1} \lambda^k \left| \nabla_{(j,t)} s(j, t-k) \right| \, \mathrm{d}k \, \mathrm{d}t \right) \mathrm{d}j$$

$$= \|w_l\|_1 \int_{\mathcal{J}} \left( \int_1^T \int_0^{T-\tau} \lambda^t \left| \nabla_{(j,t)} s(j, \tau) \right| \, \mathrm{d}t \, \mathrm{d}\tau \right) \mathrm{d}j \tag{41}$$

$$= \|w_l\|_1 \int_{\mathcal{J}} \left( \int_1^T \left( \int_0^{T-\tau} \lambda^t \, \mathrm{d}t \right) \left| \nabla_{(j,t)} s(j, \tau) \right| \, \mathrm{d}\tau \right) \mathrm{d}j$$

$$= \|w_l\|_1 \int_{\mathcal{J}} \left( \int_1^T \frac{\lambda^{T-\tau} - 1}{\ln(\lambda)} \left| \nabla_{(j,t)} s(j, \tau) \right| \, \mathrm{d}\tau \right) \mathrm{d}j$$

$$\leqslant \frac{-\|w_l\|_1}{\ln(\lambda)} \int_{\mathcal{J}} \int_1^T \left| \nabla_{(j,t)} s(j, \tau) \right| \, \mathrm{d}\tau \, \mathrm{d}j$$

$$= \|w_l\|_1 \log_\lambda(\frac{1}{e}) \int_{\mathcal{J}} \int_1^T \left| \nabla_{(j,t)} s(j, \tau) \right| \, \mathrm{d}\tau \, \mathrm{d}j.$$

The equality (40) holds because $\mathrm{d}w(i,j) = w(i,j) \, \mathrm{d}j$ as a univariate differential with fixed $i$. The underlined terms indicate the extraction of $\|w_l\|_1$. The equality (41) exploits a change of variable $\tau := t - k$, which also changes the integration interval.

Theorem 4 also holds for the discrete setting of $i$ and $t$, whose proof is similar to the above and thus omitted here. The corresponding scaling factor for the discrete setting is $\frac{\|w_l\|_1}{1-\lambda} \leqslant \|w_l\|_1 \log_\lambda(\frac{1}{e})$.

**Part (2):** For the MPPD-TV-$\ell_2$ Case,

$$\int_1^{N^l} \int_1^T |\nabla_{(i,t)} v(i,t)|^2 \, \mathrm{d}t \, \mathrm{d}i$$

$$= \int_1^T \int_1^{N^l} \left| \int_0^{t-1} \lambda^k \int_{\mathcal{J}} \nabla_{(j,t)} s(j, t-k) w_l(i,j) \, \mathrm{d}j \, \mathrm{d}k \right|^2 \mathrm{d}i \, \mathrm{d}t$$

$$= \int_1^T \int_1^{N^l} \left( \int_0^{T-\tau} \lambda^t \, \mathrm{d}t \right)^2 \left( \int_{\mathcal{J}} \nabla_{(j,t)} s(j, \tau) w_l(i,j) \, \mathrm{d}j \right)^2 \mathrm{d}i \, \mathrm{d}\tau \tag{42}$$

$$\leqslant \log_\lambda^2(\frac{1}{e}) \int_1^T \int_1^{N^l} \left( \int_{\mathcal{J}} \nabla_{(j,t)} s(j, \tau) w_l(i,j) \, \mathrm{d}j \right)^2 \mathrm{d}i \, \mathrm{d}\tau$$

$$\leqslant \log_\lambda^2(\frac{1}{e}) \int_1^T \int_1^{N^l} \left( \int_{\mathcal{J}} |\nabla_{(j,t)} s(j, \tau)|^2 \, \mathrm{d}j \right) \left( \int_{\mathcal{J}} w_l^2(i,j) \, \mathrm{d}j \right) \mathrm{d}i \, \mathrm{d}\tau \tag{43}$$

$$= \log_\lambda^2(\frac{1}{e}) \int_1^T \left( \int_{\mathcal{J}} |\nabla_{(j,t)} s(j, \tau)|^2 \, \mathrm{d}j \right) \underline{\left( \int_1^{N^l} \int_{\mathcal{J}} w_l^2(i,j) \, \mathrm{d}j \, \mathrm{d}i \right)} \mathrm{d}\tau$$

$$= \underline{\|w_l\|_F^2} \log_\lambda^2(\frac{1}{e}) \int_1^T \int_{\mathcal{J}} |\nabla_{(j,t)} s(j, \tau)|^2 \, \mathrm{d}j \, \mathrm{d}\tau.$$

The underlined terms indicate the extraction of $\|w_l\|_F^2$. The equality (42) exploits a change of variable $\tau := t - k$. The inequality (43) is derived from the Cauchy-Schwarz inequality for the $L^2$ space:

$$\left| \int_{\mathcal{J}} \nabla_{(j,t)} s(j,\tau) w_l(i,j) \, dj \right| \leqslant \left( \int_{\mathcal{J}} |\nabla_{(j,t)} s(j,\tau)|^2 \, dj \right)^{\frac{1}{2}} \left( \int_{\mathcal{J}} w_l^2(i,j) \, dj \right)^{\frac{1}{2}}. \qquad (44)$$

$\square$

## A.5 Proof of Proposition 5

*Proof.* First, we provide the definition of the Fréchet subdifferential of $f : \mathbb{R} \to \mathbb{R}$ at $w$, denoted by $\partial_w f(w)$:

**Definition 6** (The Fréchet Subdifferential)**.**

$$\partial_w f(w) := \left\{ z \in \mathbb{R} : \liminf_{\substack{u \to w \\ u \neq w}} \frac{f(u) - f(w) - z \cdot (u - w)}{\|u - w\|_2} \geqslant 0 \right\}. \qquad (45)$$

An element in the set $\partial_w f(w)$ is called a subgradient, also denoted by $\partial_w f(w)$ for simplicity. It is well-known that a subgradient of the modulus function is $\partial_w |w| = \frac{w}{|w|}$ for $w \neq 0$, or $\partial_w |w| = 0$ for $w = 0$.

As for the subgradient of MPPD-TV-$\ell_1$, it can be calculated by exploiting the Leibniz integral rule, the Fundamental Theorem of Calculus, and the chain rule for backpropagation:

$$
\begin{aligned}
&\partial_{w(i,j(i))} \left( \int_{\Theta} |\nabla_{(i,t)} v(i,t)| \, d\mu \right) \\
&= \int_{\Theta} \partial_{w(i,j(i))} \left| \sum_{k=0}^{t-1} \lambda^k \int_{\mathcal{J}(i)} \nabla_{(j,t)} s(j,t-k) \, dw(i,j(i)) \right| \, d\mu \\
&= \int_{\Theta} \mathrm{sign}(\sum_{k=0}^{t-1} \lambda^k \int_{\mathcal{J}(i)} \nabla_{(j,t)} s(j,t-k) \, dw(i,j(i))) \\
&\quad \cdot \left( \sum_{k=0}^{t-1} \lambda^k \partial_{w(i,j(i))} \left( \int_{\mathcal{J}(i)} \nabla_{(j,t)} s(j,t-k) \, dw(i,j(i)) \right) \right) \, d\mu \\
&= \int_{\Theta} \mathrm{sign}(\sum_{k=0}^{t-1} \lambda^k \int_{\mathcal{J}(i)} \nabla_{(j,t)} s(j,t-k) \, dw(i,j(i))) \cdot (\sum_{k=0}^{t-1} \lambda^k \nabla_{(j,t)} s(j,t-k)) \, d\mu. \qquad (46)
\end{aligned}
$$

It finishes the proof.

$\square$

## A.6 Additional Experimental Results

A device with an Intel(R) Xeon(R) Platinum 8352V CPU, 64-GB RAM, and an NVIDIA RTX 4090 GPU is used for CIFAR 10 and CIFAR 100, while a device with an Intel(R) Xeon(R) Gold 6348 CPU, 100-GB RAM, and an NVIDIA A800 GPU is used for Tiny ImageNet. The training times of different methods with VGG11 architecture and AT training scheme on CIFAR 10, CIFAR 100, and Tiny ImageNet data sets are provided in Table A1, which indicate that MPPD-TV-$\ell_1$ runs the fastest among the competitors. Besides, the gradient magnitudes based on the $\ell_2$ norm for different methods with WRN16 architecture and AT training scheme on Tiny ImageNet data set are provided in Figure A1, which show that MPPD-TV-$\ell_1$ converges quickly to a low gradient magnitude level around the 400-th iteration, and maintains the lowest gradient magnitude compared with MPPD-TV-$\ell_2$ and Non-MPPD. This confirms the gradient stability of MPPD-TV-$\ell_1$.

Table A1: Runtimes (in hours) of different methods with VGG11 architecture and AT training scheme on CIFAR 10, CIFAR 100, and Tiny ImageNet.

| Data Set | MPPD-TV-$\ell_1$ | MPPD-TV-$\ell_2$ | AT + FEEL | SR |
|---|---|---|---|---|
| CIFAR 10 | **9.95** | 10.01 | 13.75 | 33.89 |
| CIFAR 100 | **10.08** | 11.53 | 14.22 | 34.67 |
| Tiny ImageNet | **22.38** | 25.14 | 38.02 | 118.09 |

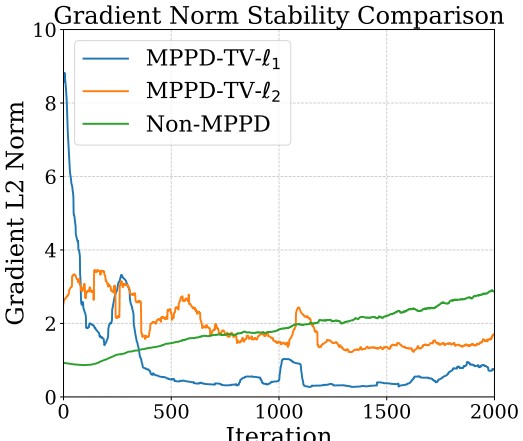

Figure A1: $\ell_2$ norms of gradients for different methods with WRN16 architecture and AT training scheme on Tiny ImageNet.

