# OpenReview forum: "A Unified Total Variation Framework for Membrane Potential Perturbation Dynamic"
_ICLR.cc/2026/Conference — ICLR 2026 Poster_

### Official Review · Reviewer_1NUH · 2025-10-16

**Soundness:** 3
**Presentation:** 2
**Contribution:** 2
**Rating:** 6
**Confidence:** 4

**Summary:**

This paper reframes a heuristic SNN stabilization mechanism (MPPD) into a rigorous TV theory, generalizes it via a new ℓ₁-based variational framework, and validates its advantage in both adversarial and noisy environments.

#### **1. Research Problem**

* Spiking Neural Networks (SNNs) are vulnerable to adversarial and noisy perturbations that destabilize their dynamics. The existing *Membrane Potential Perturbation Dynamic* (MPPD) technique empirically improves robustness but lacks solid theoretical grounding.
* The paper tries to reveal mathematical nature of MPPD, and how can it be formalized to enhance SNN robustness in a principled way.

#### **2. Proposed Method**

* The authors prove that MPPD is mathematically equivalent to Total Variation (TV)
* Based on this equivalence, they introduce the MPPD–TV–ℓ₁ framework, extending prior ℓ₂-based formulations (MPPD–TV–ℓ₂).


#### **3. Theoretical Contributions**

* Rigorous proof that MPPD is TV under measurable perturbations.
* Establishment of a new TV–ℓ₁ regularization theory for SNNs, encompassing:

  * The coarea formula specific to SNN membrane dynamics.
  * A dominated TV property showing layer-wise boundedness of perturbations.
  * A closed-form subgradient that enables backpropagation through non-smooth TV terms.

#### **4. Experimental Contributions**

* Across CIFAR-10 and CIFAR-100, the MPPD–TV–ℓ₁ model outperforms both the ℓ₂ variant and other baselines under Gaussian noise and adversarial attacks (FGSM, PGD, CW, AutoAttack).
* Demonstrates higher resistance to increased attack intensity and step size, confirming TV–ℓ₁’s superior denoising behavior.

**Strengths:**

### **1. Theoretical Contributions**
Overall, this paper provides a theoretically sound bridge between signal variation analysis and adversarial robustness in SNNs, offering a new mathematical perspective for neuromorphic robustness theory with clear derivations.

* Introduces a novel reinterpretation of membrane potential perturbation dynamics (MPPD) as a form of total variation (TV), unifying biological spiking dynamics and variational regularization theory in an original analytical framework.
* Transforms prior MS-MPPD regularization (previously heuristic) into a mathematically grounded TV–ℓ₂ model and further generalizes it into the TV–ℓ₁ formulation, expanding the functional space of admissible membrane potentials and enabling sharper perturbation modeling.
* Mathematical Rigorly establishes formal results including the Coarea formula for spiking potentials, the Dominated TV Property linking layerwise stability to weight norms, and a closed-form subgradient for optimization without additional computational cost.



### **2. Experimental Contributions**
Experimental result supports that total variation regularization can serve as a universal principle for temporal–spatial robustness in SNNs.

* Conducts extensive controlled experiments on CIFAR-10 and CIFAR-100 using both VGG11 and WRN16 backbones under Gaussian and adversarial training, comparing six state-of-the-art SNN methods.
* Demonstrates consistent and often substantial gains in adversarial accuracy, validating that MPPD-TV–ℓ₁ effectively suppresses perturbations across attack intensities and steps.
* Shows that the closed-form subgradient introduces no measurable computational overhead while maintaining compatibility with mainstream deep learning frameworks.

**Weaknesses:**

Overall, this paper provides rigorous theoretical analysis and effective experimental results to demonstrate the theoretical foundations of MPPD and offer a more complete version. I particularly appreciate the paper's rigorous treatment of pulse discontinuities, which is rare but meaningful in the SNN field.

1. My main concern is that MPPD has not yet become a mainstream method for SNNs, and this paper is almost entirely based on this premise, which limits its broader impact. I am not sure how interested most people in SNN field are in it. For example, can the techniques for handling spike discontinuities in the paper be generalized to other SNN research?

2. Please add some missing citations. Some related work also focuses on smoothing membrane potential perturbations under adversarial attacks with uses different methods, such as dynamic thresholding (https://arxiv.org/pdf/2308.10373) and stochastic gating (https://ojs.aaai.org/index.php/AAAI/article/view/27804).

3. In Theorem 4, "assume every node i in layer L uses the same set of preceding nodes in layer L-1" which does not hold for sparse, or skip-connected SNN architectures. Relaxing this constraint would enhance generality.

**Questions:**

1.Can the techniques for handling spike discontinuities in the paper be generalized to other SNN research?
2.Please add some missing citations.

---

> ### Author Response · Authors · 2025-11-17
>
> We thank the reviewer for the highly encouraging summary, which accurately highlights the significance of connecting the Membrane Potential Perturbation Dynamics (MPPD) to the rigorous theory of Total Variation (TV). We are particularly glad that the reviewer appreciates the novel handling of spike discontinuities, which is a central technical challenge in SNNs.
>
> **Q1.** While MPPD serves as the initial motivation, the paper's central contribution is the TV framework itself, which is applicable to a far wider range of SNN models than the original MPPD heuristic.
>
> MPPD-TV-$\ell_1$ provides a principled, theoretically-grounded method to regularize the spatio-temporal dynamics of most SNNs, regardless of their specific architectural components (like dynamic thresholds or stochastic gating). The TV-$\ell_1$ regularizer penalizes the instability of the internal membrane potential, offering a fundamental approach to robustness.
>
> The technique for handling spike discontinuities is highly generalizable. The closed-form subgradient (Proposition 5) is derived specifically to address the non-smoothness for spike generation in most SNNs (LIF, IF, etc.). Our derivation provides a mathematically sound, closed-form subgradient for a TV term based on this discontinuity. This technique is applicable to most SNN architectures where a TV term is used to stabilize layer-wise internal state, directly addressing a core challenge in SNN training. We emphasize this broader significance in the revised introduction and conclusion.
>
> **Q2.** Thanks for raising these references and we have added them in the revised manuscript.
>
> **Replies to Other Concerns in Weaknesses.**
>
> **W3.** The constraint was introduced primarily to simplify notation and ease comprehension. It can be easily relaxed to general situations, including sparse or skip-connected SNN architectures. For such architectures, the weight matrix $w_l$ simply has zero entries corresponding to non-connections, and the set of preceding nodes $J(i)$ is a proper subset of the entire layer $(l-1)$. The proof structure and the fundamental dominating bound remain valid. We have added an explanation under Theorem 4 in the revised manuscript.

---

> ### Author Response · Authors · 2025-11-27
>
> Dear reviewer 1NUH,
>
> We have carefully considered all the valuable points and concerns you raised in your initial review. As the deadline for discussion is approaching, we respectfully ask whether our response has addressed your concerns, or you still have questions to raise. Your comments are highly valuable to us. Thank you once again for your professional time and insightful effort in reviewing our submission.
>
> Best regards,
>
> Authors of Submission 11931

---

### Official Review · Reviewer_sYtm · 2025-10-23

**Soundness:** 4
**Presentation:** 2
**Contribution:** 2
**Rating:** 4
**Confidence:** 4

**Summary:**

This work is build on the concept of Membrane Potential Perturbation Dynamics (MPPD) as a method for enhancing the robustness SNNs, particularly in the face of adversarial perturbations. The authors propose that MPPD can be framed as a Total Variation (TV) model and further develop a novel MPPD-TV-L1 framework, which they show improves the robustness of SNNs in adversarial environments. The proposed approach demonstrates superior performance over existing TV-L2 models on image classification tasks using the CIFAR-10 and CIFAR-100 datasets.

**Strengths:**

1: This work proves that MPPD is equivalent to TV, providing a strong mathematical foundation that underpins the proposed method.

2: The experimental setup is comprehensive, involving state-of-the-art methods and adversarial training schemes.

3: The motivation that extend the existing TV-L2 framework to TV-L1 is well-articulated.

4: The proposed framework has clear practical implications for improving the security and reliability of SNNs.

**Weaknesses:**

A major concern is the incremental novelty of this work. The MPPD-TV-L2 framework was already proposed in Ding et al. (2024), and this work introduces the MPPD-TV-ℓ1 framework. Furthermore, as shown in Figure 1, in the case of AT+Reg, the MPPD-TV-ℓ1 shows only minor (or no) improvement. This suggests that MPPD-TV-ℓ1 has a similar effect to the regularizer (Ding et al., 2022), which somewhat weakens the novelty and necessity of this work.

The writing and clarity of the paper can be improved. For example, the title may mislead readers into thinking that the paper proposes MPPD (which it does not). Additionally, there is no punctuation for $\epsilon$ in Equation (3), and the term "MS-MPPD" looks awkward in Equations (4) and (5).

**Questions:**

See the weaknesses section for my major concerns.

In Table 1, the cases where MPPD-TV-L1 performs worse than MPPD-TV-L2 all occur with the FGSM and the APGD_DLR. It would be helpful if the authors could provide a theoretical or intuitive explanation for this behavior.

Could you include a comparison that shows the performance of ANNs in handling adversarial perturbations, to better highlight the relative robustness (if any) of SNNs?

---

> ### Author Response · Authors · 2025-11-17
>
> We sincerely thank the reviewer for the thoughtful summary and constructive feedback, especially for recognizing the strong mathematical foundation and the comprehensive experimental setup.
>
> **Replies to Weaknesses.**
>
> **W1.** We respectfully disagree that the novelty is incremental, and we believe the behavior in the AT+Reg case actually **highlights** the significance of our method.
>
> The extension from $\ell_2$ to $\ell_1$ is not only a formula change. In fact, we aim to unify both $\ell_2$ and $\ell_1$ regularizations into the fundamental TV regularization mechanism, in order to reveal the working mechanism of such SNN-based models in capturing and suppressing adversarial perturbations and shed light on the essence of perturbation characterization.
>
> Note that AT+Reg is already a heavy double penalization for non-robust dynamics. The fact that MPPD-TV-$\ell_1$ shows little further improvement under AT+Reg suggests that MPPD-TV-$\ell_1$ is implicitly achieving the desired robust regularization effect that the explicit Reg treatment of ref. (Ding et al., 2022) is designed for. In the more common and important training scenario AT, MPPD-TV-$\ell_1$ consistently shows better performance, which proves its practical necessity and advantage as a standalone, effective robust training method. We add an explanation in Section 4.2 of the revised manuscript.
>
> **W2.** We have changed the title to "A Unified Total Variation Framework for Membrane Potential Perturbation Dynamic", in order to highlight the TV formulation of MPPD, which belongs to our main contribution. We have also changed the $MS-MPPD$ to $\mathcal{L}_{MS\text{-}MPPD}$ throughout the paper, to make a better appearance. As for $\epsilon$, it is a function of $t$ and thus we just keep the current appearance. We have revised the manuscript to further improve writing and clarity.
>
> **Replies to the Rest Questions.**
>
> **Q2.** As explained above, MPPD-TV-$\ell_1$ is implicitly achieving the desired robust regularization effect that the explicit Reg treatment is designed for, thus MPPD-TV-$\ell_1$ shows little further improvement over MPPD-TV-$\ell_2$ under AT+Reg. FGSM is a foundational adversarial attack method, the predecessor of PGD and APGD. Thus Reg is more effective on FGSM than on PGD, and MPPD-TV-$\ell_1$ is not so effective in the FGSM+AT+Reg case. However, PGD is a stronger attack than FGSM, and MPPD-TV-$\ell_1$ is effective in all the PGD+AT+Reg case. As for APGD_DLR, MPPD-TV-$\ell_1$ remains within a $1\%$ accuracy gap relative to MPPD-TV-$\ell_2$ in the three underperforming cases, which indicates an even match. Moreover, since the task is a classification, APGD_CE is more effective than APGD_DLR because the CE (cross entropy) determines the final classification results.
>
> **Q3.** We add experimental results of two typical ANNs: ANN-PGD-AT (Madry et al., 2018) and ANN-RiFT (Zhu et al., 2023), in Tables 1 and 2 of the revised manuscript. MPPD-TV-$\ell_1$ outperforms these two ANNs in most cases, which shows its better adversarial robustness.

---

> > ### Comment · Reviewer_sYtm · 2025-11-26
> >
> > The authors have addressed most of my concerns in the rebuttal and the revised manuscript. I have therefore raised my score.

---

> > > ### Author Response · Authors · 2025-11-27
> > >
> > > Dear reviewer sYtm,
> > >
> > > Thank you once again for your professional time, insightful effort, and the careful consideration of our response. We are pleased that our revisions have addressed your concerns.
> > >
> > > Best regards,
> > >
> > > Authors of Submission 11931

---

### Official Review · Reviewer_mt4t · 2025-10-31

**Soundness:** 3
**Presentation:** 2
**Contribution:** 2
**Rating:** 4
**Confidence:** 4

**Summary:**

The paper presents a theoretical analysis of the Membrane Potential Perturbation Dynamic (MPPD) in Spiking Neural Networks (SNNs). The authors' main contribution is framing MPPD as a form of Total Variation. This provides a theoretical foundation for MPPD.

**Strengths:**

The primary strength is the formal theoretical analysis offered for the MPPD approach, addressing a known gap in the field.
​​
The method appears to achieve good performance, suggesting the theoretical insight translates into practical benefits.

**Weaknesses:**

A major issue is that Table 1 does not report the "clean" accuracy (performance without noise), making it impossible to evaluate the true cost of the denoising improvement.

​​The choice of the key parameter ζ in Section 4.1 is not explained. It is unclear if it was set arbitrarily, tuned for this work, or copied from another paper. If compared papers in Table 1 used different ζvalues, the comparison is misleading and should be noted.

The preliminary discussion describes previous MPPD work in a discrete setting, but the proposed method uses a continuous formulation. The paper does not justify this shift or explain how the continuous form is compatible with or translates to the discrete SNN simulation.

The proposed TV loss might not fully capture the original MPPD behavior. Specifically, when a reset mechanism is involved, small perturbations that are insufficient to evoke a spike may be excluded from the loss calculation, potentially making the model less sensitive to certain types of noise.

​​In Equation 2.4, using a shorter minus sign (e.g., \text{-}) would improve visual alignment and readability.

​​The statement that previous work "lacks reliable explanations and theoretical foundation" (Line 55) is too strong, especially if the authors' own prior work is in the same theoretical domain. This should be phrased more precisely.

**Questions:**

Why was the continuous formulation chosen for the proposed method when the preliminary MPPD description is discrete? How is the continuous formulation implemented or made compatible with the discrete-time dynamics of an actual SNN?

What are the clean (noise-free) accuracy scores corresponding to the results in Table 1? This is critical for assessing the performance-robustness trade-off.

How does the proposed TV loss account for sub-threshold membrane potential perturbations that do not lead to a spike reset? Does excluding these perturbations limit the model's sensitivity to low-intensity noise?

Minor: The use of bold font in the main text seems arbitrary and should be applied more consistently.

---

> ### Author Response · Authors · 2025-11-17
>
> We sincerely thank the reviewer for the thorough and constructive review. We are pleased that the reviewer recognizes the primary strength of our work: providing a formal theoretical analysis for the Membrane Potential Perturbation Dynamic (MPPD) approach by framing it as a form of Total Variation (TV).
>
> **Q1.** We must emphasize that all the theoretical and technical results of this paper (e.g., Theorems 1,2,3,4) already **hold for both continuous and discrete settings**, including all the proofs in the appendices. For example, Theorem 2 presents the coarea formula for both continuous and discrete settings. We have also provided a thorough theoretical proof regarding both continuous and discrete settings in Appendix A.2, where Part (1) and Part (2) correspond to the discrete and the continuous settings, respectively. Additionally, we provide the scaling factor for the discrete setting of Theorem 4: $\frac{\\|w_l\\|_1}{1-\lambda}$.
>
> The key difference between the continuous and discrete settings lies in whether we use the Lebesgue measure or the discrete counting measure to calculate the TV, respectively. As for practical discrete implementation, it can be integrated into the standard discrete-time backpropagation process of the SNN.
>
> **Q2.** We add clean accuracy experiments of different methods in Tables 1 and 2 of the revised manuscript. MPPD-TV-$\ell_1$ outperforms MPPD-TV-$\ell_2$ and Non-MPPD on both clean and perturbed data. This indicates that MPPD-TV-$\ell_1$ really improves robustness not just against adversarial perturbations, but also against other types of detrimental noise.
>
> **Q3.** The proposed MPPD-TV-$\ell_1$ accounts for sub-threshold perturbations effectively by acting as a regularizer on the weight landscape in the evolution of SNN:
>
> The TV formulation penalizes the total accumulation of potential changes over time ($\int |\nabla_{(\cdot,t)}v(\cdot,t)| d\mu$, see Theorem 3), not just the potential at the moment of a spike. The membrane potential $v(i,t)$ evolves continuously based on input currents, even when no spike occurs.
>
> The gradient of TV with respect to the weights, $\partial_w TV$ in Proposition 5, captures the sensitivity of the TV to the weights at every timestep $t$, regardless of whether the potential crosses the threshold and is reset.
>
> By minimizing the TV, the model weights are enforced to produce membrane potential trajectories that are globally smoother and less responsive to small changes in the input (noise). This inherent smoothness regulates the weight updates such that even small, sub-threshold input perturbations are suppressed by a less-sensitive weight profile.
>
> Therefore, MPPD-TV-$\ell_1$ successfully enforces robustness even in the sub-threshold regime by regularizing the influence of weights on the membrane potential's overall trajectory. We have strengthened such explanations in the corresponding parts below Theorem 3 and Proposition 5 of the revised manuscript.
>
> **Q4.** We have changed some bold-font content to plain text in the revised manuscript.
>
> **Replies to Other Concerns in Weaknesses.**
>
> **W2.** All the settings of training and test including $\zeta$ strictly follow those of ref. (Ding et al., 2024) to make fair comparisons. Although according to the rule of ref. (Ding et al., 2024), different training perturbations $\zeta$ (i.e., $10/255$, $6/255$, $7/255$) are set according to different adversarial training schemes (i.e., Gaussian noise, AT, AT+Reg), the same $\zeta$ is uniformly used for all the compared methods in each scenario to ensure fairness. Different training perturbations are used because different adversarial training schemes have different sensitivities to the perturbation strength.
>
> **W5. \& W6.** Thanks for these good suggestions. We have shortened this minus sign throughout the revised manuscript, and rephrased that the previous work may be insufficient in perturbation characterization.

---

> ### Author Response · Authors · 2025-11-27
>
> Dear reviewer mt4t,
>
> We have carefully considered all the valuable points and concerns you raised in your initial review. As the deadline for discussion is approaching, we respectfully ask whether our response has addressed your concerns, or you still have questions to raise. Your comments are highly valuable to us. Thank you once again for your professional time and insightful effort in reviewing our submission.
>
> Best regards,
>
> Authors of Submission 11931

---

### Official Review · Reviewer_X9cd · 2025-10-31

**Soundness:** 3
**Presentation:** 3
**Contribution:** 2
**Rating:** 6
**Confidence:** 4

**Summary:**

The paper establishes a theoretical foundation for Membrane Potential Perturbation Dynamics (MPPD) in spiking neural networks (SNNs), proving that MPPD corresponds to Total Variation. The authors propose a new framework that improves robustness to adversarial perturbations compared to the existing MPPD model. Experimental results on CIFAR-10 and CIFAR-100 demonstrate that MPPD achieves superior accuracy and robustness under various adversarial attacks.

**Strengths:**

- The paper provides a clear mathematical link between MPPD and total variation, offering the first formal theoretical explanation for an empirically effective mechanism in SNN robustness.
- Experimental results on CIFAR-10 and CIFAR-100 demonstrate that MPPD achieves superior accuracy and robustness under various adversarial attacks.

**Weaknesses:**

- The experiments are limited to CIFAR-10 and CIFAR-100. These are small-scale image datasets. Experiments on more large-scale datasets and neuromorphic datasets are encouraged.
- Although the paper mentions efficiency, there is no detailed analysis of training time and gradient stability.
- The paper does not report clean test accuracy alongside adversarial robustness results.

**Questions:**

In the abstract, the authors state that “this finding may provide a new insight into the essence of perturbation characterization.” Could the authors clarify what specific insights are being referred to here?

---

> ### Author Response · Authors · 2025-11-17
>
> We sincerely thank the reviewer for the careful review of our manuscript and for providing valuable and constructive feedback. We greatly appreciate the recognition of our core contribution: the first-ever theoretical establishment of the mathematical connection between Membrane Potential Perturbation Dynamics (MPPD) and Total Variation (TV), and the introduction of the MPPD-TV-$\ell_1$ framework with superior performance.
>
> **Q1.** We refer to the first formal mathematical elevation of MPPD, an empirically effective SNN robustness mechanism, to the rigorous theoretical level of TV. The specific insights are:
>
> Theoretical Foundation: MPPD is no longer a heuristic regularizer, but a TV-$\ell_2$ framework with solid theoretical underpinnings.
>
> New Robustness Mechanism: Based on the coarea formula (Theorem 2), we found that TV-$\ell_1$ offers a stronger de-noising advantage, providing a new perspective on the nature of perturbation characterization in SNNs.
>
> **Replies to Other Concerns in Weaknesses.**
>
> **W1.** We add experimental results on Tiny ImageNet data set [a] in Table 2 of the revised manuscript. Tiny ImageNet is a large-scale data set with $500$ $64\times 64$ downsized images for each of the $200$ classes. Results show that MPPD-TV-$\ell_1$ achieves the best performance among the competitors in most cases. Hence MPPD-TV-$\ell_1$ is also effective on large-scale data.
>
> **W2.** We supplement experimental results on training time and gradient stability in Appendix A.6, Table A1, and Figure A1. Results show that MPPD-TV-$\ell_1$ runs the fastest among the competitors. Besides, MPPD-TV-$\ell_1$ converges quickly to a low gradient magnitude level around the $400$-th iteration, and maintains the lowest gradient magnitude compared with MPPD-TV-$\ell_2$ and Non-MPPD. This confirms the gradient stability of MPPD-TV-$\ell_1$.
>
> **W3.** We add clean accuracy experiments of different methods in Tables 1 and 2 of the revised manuscript. MPPD-TV-$\ell_1$ outperforms MPPD-TV-$\ell_2$ and Non-MPPD on both clean and perturbed data. This indicates that MPPD-TV-$\ell_1$ really improves robustness not just against adversarial perturbations, but also against other types of detrimental noise.
>
> [a] Le, Ya, and Xuan Yang (2015). Tiny ImageNet visual recognition challenge. CS 231N, 7(7), 3.

---

> ### Author Response · Authors · 2025-11-27
>
> Dear reviewer X9cd,
>
> We have carefully considered all the valuable points and concerns you raised in your initial review. As the deadline for discussion is approaching, we respectfully ask whether our response has addressed your concerns, or you still have questions to raise. Your comments are highly valuable to us. Thank you once again for your professional time and insightful effort in reviewing our submission.
>
> Best regards,
>
> Authors of Submission 11931

---

### Author Response · Authors · 2025-11-30
**Summary of Discussions [2/2]**

## Addressing Experimental Concerns ##

We have addressed the major concerns regarding the scope and completeness of our experimental results:

* **Large-Scale Data Set:** We supplement experiments on the large-scale **Tiny ImageNet** data set (shown in Table 2), where MPPD-TV-$\ell_1$ achieves the **best performance** among competitors in most cases. This confirms the effectiveness of MPPD-TV-$\ell_1$ on large-scale data.

* **Clean Test Accuracy:** We add the **clean (noise-free) accuracy scores** to the revised Tables 1 and 2. Results show that MPPD-TV-$\ell_1$ outperforms other competitors in most cases of **both clean and perturbed data**, which indicates a substantial improvement in robustness.

* **Efficiency Analysis:** We include a detailed analysis of **training time and gradient stability** in Appendix A.6, Table A1, and Figure A1. The results show that MPPD-TV-$\ell_1$ runs the **fastest** and maintains the **lowest gradient magnitude**, which confirms its stability.

* **Comparison with ANNs:** We add experimental results that compare MPPD-TV-$\ell_1$ with two typical ANNs (ANN-PGD-AT and ANN-RiFT) to the revised Tables 1 and 2. **MPPD-TV-$\ell_1$ outperforms these ANNs** in most cases, which shows its superior adversarial robustness.

* **Novelty of MPPD-TV-$\ell_1$:** We emphasize that the **extension from $\ell_2$ to $\ell_1$ is fundamental**, which aims to unify both regularizations within the TV characterization. The good performance of MPPD-TV-$\ell_1$ in the common Adversarial Training (AT) scenario proves its practical necessity and advantage as a standalone robust training method. [`sYtm`: _The authors have addressed most of my concerns in the rebuttal and the revised manuscript. I have therefore raised my score_.]

* **Choice of Perturbation Budget $\zeta$:** All the settings of training and test including $\zeta$ strictly follow those of ref. (Ding et al., 2024) to make fair comparisons. The **same $\zeta$ is uniformly used for all the compared methods** in each scenario to ensure fairness, which is a **fundamental research criterion**.

## Manuscript Revisions ##

We have made several revisions to improve the clarity and rigor of the manuscript:

* **Title Change:** We have revised the title to "A Unified Total Variation Framework for Membrane Potential Perturbation Dynamic" to better reflect our main contribution.

* **Citation and Constraint Relaxation:** We add the requested missing citations to related work. We also add an explanation to show how the **constraint in Theorem 4 can be relaxed for sparse or skip-connected architectures** by simply having zero entries in the weight matrix.

* **Formatting:** We correct the minus sign formatting and revise some arbitrary bold text for consistency.

## Addressing Reviewer `mt4t`'s Initial Rating ##

The core issues raised by reviewer `mt4t` are primarily based on **information gaps and clarity**: the lack of **clean accuracy**, the choice of **perturbation strength $\zeta$**, the compatibility of the **continuous/discrete formulation**, and the handling of **sub-threshold perturbations**. Our responses systematically address all of these points, which **fill the information gaps** that cause the initial marginal score.

* **Clean Accuracy (Q2/W1):** We add clean accuracy results to the revised Tables 1 and 2, which show that **MPPD-TV-$\ell_1$ outperforms MPPD-TV-$\ell_2$ and Non-MPPD on both clean and perturbed data**. This addresses the critical concern about the performance-robustness trade-off.

* **Continuous/Discrete Formulation (Q1/W3):** We clarify that the theoretical results **hold for both continuous and discrete settings** (See all the theorems in the main text and the proofs in Appendix A). This directly justifies the continuous formulation and explains its compatibility with discrete SNN dynamics.

* **Sub-Threshold Perturbations (Q3/W4):** We provide a detailed explanation to emphasize that the TV formulation penalizes the total accumulation of potential changes **over time (Theorem 3 and the paragraph below)**, not just the potential at the moment of a spike. The gradient of TV **captures sensitivity at every timestep (Proposition 5 and the paragraph below)**, which effectively enforces robustness in the sub-threshold regime.

* **Choice of Perturbation Budget $\zeta$ (W2):** We clarify that all the parameter settings including $\zeta$ strictly follow ref. (Ding et al., 2024) and are the same to all the compared methods, in order to **ensure fair comparisons**.

---

### Author Response · Authors · 2025-11-30
**Summary of Discussions [1/2]**

We are grateful for the thorough, constructive, and insightful reviews of our manuscript. We are pleased that the reviewers (`X9cd, mt4t, sYtm, 1NUH`) recognize our primary contribution: **the first formal theoretical establishment** of the mathematical connection between **Membrane Potential Perturbation Dynamics (MPPD)** and **Total Variation (TV)**, and the introduction of the superior **MPPD-TV-$\ell_1$ framework**.

# Reviewer `sYtm`'s Positive Feedback with Valid Score Raise (Newest 6,4,6,6, Average: 5.5) #

We would like to draw attention to the final comment from reviewer `sYtm`, who explicitly states: **"The authors have addressed most of my concerns in the rebuttal and the revised manuscript. I have therefore raised my score", on 17:49, 25 Nov 2025 AOE, which is two days ahead of the reviewer identity leak. Hence this is a valid score raise (from 4 to 6)**. It indicates that the effectiveness of our revisions and the successful resolution of all major concerns raised by reviewer `sYtm` regarding the incremental novelty and writing clarity. We are pleased that our efforts have led to a stronger assessment of our work.

Taking this valid score raise into consideration, **the newest valid scores for this paper are 6,4,6,6, having an average of 5.5**.

## Key Insights \& Theoretical Clarity ##

The main theoretical insight is the formal elevation of MPPD from a heuristic regularizer to a **TV-based framework with solid theoretical underpinnings**. This mathematical link (MPPD $\leftrightarrow$ TV) unifies biological spiking dynamics and variational regularization theory, and provides a principled method for **perturbation characterization** in Spiking Neural Networks (SNNs).

* **Continuous vs. Discrete Settings:** We clarify that all theoretical and technical results hold for **both continuous and discrete settings**, including the coarea formula (Theorem 2) and all the proofs in Appendix. The practical discrete implementation can be integrated into the standard discrete-time backpropagation of the SNN.

* **Handling Sub-Threshold Perturbations:** The proposed MPPD-TV-$\ell_1$ effectively accounts for **sub-threshold perturbations**. The TV formulation acts as a **regularizer on the weight landscape** by penalizing the total accumulation of potential changes over time. The gradient of TV captures the sensitivity of weights at **every timestep**, ensuring that even small and sub-threshold input perturbations are suppressed by a smoother and less-sensitive weight profile.

* **Generalizability of Techniques:** The closed-form subgradient derived for the TV term (Proposition 5), which addresses the non-smoothness of spike generation, is **highly generalizable** to most SNN architectures (e.g., LIF, IF) where a TV term is used to stabilize the internal state.

---

### Meta-Review · Area_Chair_JKSx · 2026-01-06

**Summary:**

This paper establishes a formal theoretical connection between Membrane Potential Perturbation Dynamics and Total Variation, reframing a previously heuristic robustness technique for spiking neural networks within a rigorous variational framework. Reviewers generally agree that the theoretical contribution is solid and that the empirical results support the proposed framework, while differing in their assessment of the work’s novelty and breadth of impact.

**Reviewer Concerns:**

Most of concerns were addressed in the rebuttal and revised manuscript through added experiments on larger datasets, clean accuracy reporting, efficiency and gradient stability analysis, and clearer theoretical explanations.

**Reviewer Scores:**

Reviewer scores are moderately positive but mixed. Two reviewers rated the paper marginally above the acceptance threshold, emphasizing the strength of the theoretical analysis and improved experimental support after revision. Other reviewers remained at borderline or marginal reject, citing concerns about novelty and general interest. One reviewer explicitly raised the score after the rebuttal, indicating that several major concerns were satisfactorily resolved.

---

### Decision · Program_Chairs · 2026-01-26

Accept (Poster)